# M-Star: Markovian Projection of Star-Shaped Diffusion for Exponential Family Distributions

François Bertholom [* 1]   Khalid Oublal [* 1 2]

## Abstract

Diffusion models achieve state-of-the-art performance in generative modeling but are limited by their reliance on Gaussian noise and the high computational cost of iterative sampling. Star-shaped diffusion addresses the former by introducing a non-Markovian forward process, yet this comes at the expense of temporal coherence in the reverse process. We propose a novel framework that resolves this trade-off by learning a Markovian projection of a star-shaped forward process, and its reversal. This design enables learning over a broad class of exponential models and recovers DDPM as a special case. It is particularly well-suited for knowledge distillation, allowing few-step or even single-step generation. Experiments demonstrate the effectiveness and flexibility of our approach across multiple generative tasks. Code and demo are available at: https://oublalkhalid.github.io/MStar-Diffusion/.

## 1. Introduction

Diffusion models (Sohl-Dickstein et al., 2015; Ho et al., 2020; Song et al., 2020b) are the undisputed state-of-the-art in generative modeling, however this success is largely built on the convenient but restrictive assumption of Gaussian noise. While effective for various tasks, it becomes problematic for certain real-world data structures, e.g., constrained geometries or directional data, that may arise in specific settings like climate science. Tentatives to extend the framework of Denoising Diffusion Probabilistic Models (DDPM) (Ho et al., 2020) to these non-Gaussian domains has resulted in a fragmented landscape, requiring complex derivations for each new distribution (e.g., Beta (Zhou et al.,

*Equal contribution [1]Institut Polytechnique de Paris, Palaiseau, France [2]LTCI, Télécom Paris, Palaiseau, France. Correspondence to: Khalid Oublal <khalid.oublal@ip-paris.fr>, François Bertholom <francois.bertholom@telecom-sudparis.eu>.

*Proceedings of the 43$^{rd}$ International Conference on Machine Learning*, Seoul, South Korea. PMLR 306, 2026. Copyright 2026 by the author(s).

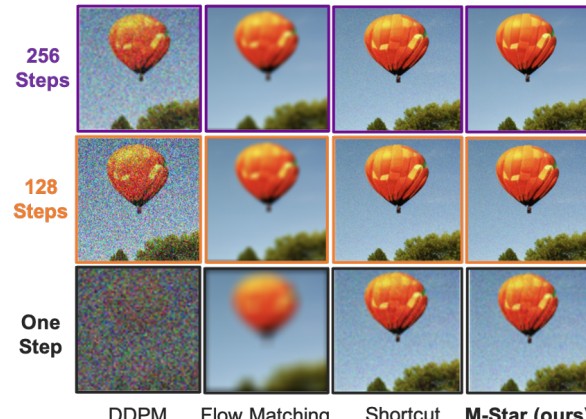

*Figure 1.* Comparison of image quality produced by (left to right) DDPM (Ho et al., 2020), Flow Matching (Lipman et al., 2022), Shortcut (Frans et al., 2025), and M-Star across different numbers of sampling steps. All models were trained and tested on ImageNet (Deng et al., 2009). Learning to predict the clean data from any timestep makes M-Star well-suited for distillation.

2023), Dirichlet (Avdeyev et al., 2023), or von Mises-Fisher (Dosi et al., 2025)), rather than a unified theory.

To overcome this rigidity, Star-Shaped Denoising Diffusion Probabilistic Models (SS-DDPM) (Okhotin et al., 2023) replace the Markovian forward process of DDPMs with a non-Markovian "hub-and-spoke" architecture. Noisy samples are conditioned directly on the clean data, and a cleverly-defined sufficient tail statistic allows the efficient implementation of the reverse process. While this design is compatible with a broad class of exponential family distributions, it introduces a critical flaw: temporal incoherence. Since the noisy states are conditionally independent given the data, the reverse process may fail to preserve the context of the previously generated states. High variance in clean data prediction is particularly detrimental, as it not only degrades final sample quality but also hinders the adoption of acceleration techniques such as progressive distillation.

To address this challenge, we introduce the Markovian-Projected Star-Shaped Diffusion (M-Star) framework. Our approach combines the flexibility of star-shaped forward processes with the stability of DDPMs. Specifically, we introduce a Markovian projection that exactly mimics the marginal distributions of a given star-shaped forward pro-

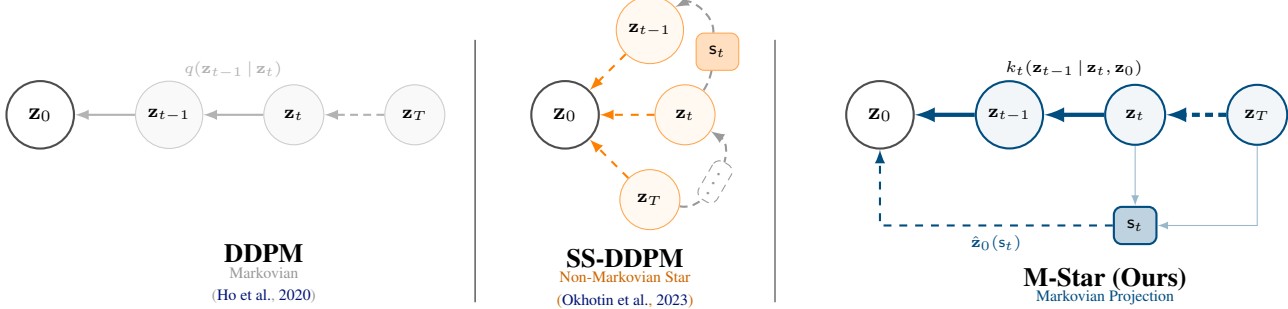

*Figure 2.* (Left) DDPM relies on a Markovian reverse process, its formulation makes it limited to Gaussian distributions. (Center) SS-DDPM uses a star-shaped graph where noisy states are conditionally independent given $\mathbf{z}_0$. This framework is flexible but the lack of sequential dependencies leads to temporal incoherence during sampling. (Right) M-Star reintroduces a Markovian structure from a star-shaped graph via Markovian Projection. This restores path consistency while retaining flexibility.

cess, and learn the reversal of this projection.

**Contributions.** We propose M-Star, a novel diffusion framework that leverages Markovian projections of star-shaped forward processes and recovers standard DDPMs as a special case. We provide principled guidelines to parameterize M-Star models, in particular we show how robust noise schedules from the Gaussian DDPM literature can be transferred to M-Star. Lastly, we show that M-Star is highly amenable to progressive distillation, validating its capability for high-quality few-step generation.

**Outline.** We start by reviewing related works in Section 2. In Section 3, we recall some basics on DDPMs and star-shaped diffusion. We introduce M-Star in Section 4, where we also discuss parameterization and distillation. Section 5 is dedicated to experiments highlighting the competitive performance of the proposed method.

## 2. Related Works

Recent works have extended diffusion to non-Gaussian distributions. Examples include Binomial (Sohl-Dickstein et al., 2015), Multinomial (Hoogeboom et al., 2021), and Gamma diffusion (Nachmani et al., 2021). However, these approaches require specific derivations for each new distribution, lacking a unified framework. Alternative approaches leverage Riemannian Score Matching (De Bortoli et al., 2022; Huang et al., 2022) or Flow Matching (Lipman et al., 2022; Chen & Lipman, 2023) to handle data on manifolds. Meanwhile, exponential family distributions have been used to solve specific tasks (Guzmán-Cordero et al., 2025; Micheli et al., 2025), and specialized objectives were developed for discrete data (Lou et al., 2023; Zhang et al., 2025). Cold Diffusion (Bansal et al., 2023) introduces a deterministic reversal process for arbitrary degradation, and Shortcut models (Frans et al., 2025) condition the network on the number of steps. In comparison, our work remains anchored in the core principles of diffusion models. Building upon star-shaped DDPMs (Okhotin et al., 2023), it provides

a general framework to learn diffusion models over a wide range of exponential family distributions.

Another major bottleneck for diffusion models is their slow iterative sampling. Denoising Diffusion Implicit Models (DDIMs) (Song et al., 2020a) enable deterministic acceleration by introducing a non-Markovian forward process. In discrete state spaces, ReMDM (Lou et al., 2023) plays an analogous role through a bridge-style reparameterization. M-Star generalizes neither: it instead constructs a Markovian projection of a star-shaped forward process directly in the natural-parameter space of an arbitrary exponential family. Subsequently, Progressive Distillation (Salimans & Ho, 2022) established a standard for compressing multi-step samplers into few-step models. This has paved the way for more recent optimizations (Yin et al., 2024a;b; Xu et al., 2025), and for the development of consistency models (Song et al., 2023) as well as general acceleration frameworks (Chadebec et al., 2025). Our framework aligns closely with these goals: training the model to predict the clean data from each step makes it better-suited for progressive distillation.

## 3. Background on Diffusion Frameworks

### 3.1. Denoising Diffusion Probabilistic Models

We briefly review the DDPM framework. Consider the task of sampling from a target distribution $q(\mathbf{z}_0)$, for which we only have access to a finite set of observations. A diffusion model defines a fixed forward process $q(\mathbf{z}_{1:T} \,|\, \mathbf{z}_0)$ that gradually adds noise to data until the final state becomes independent of the initial input, i.e., $q(\mathbf{z}_T \,|\, \mathbf{z}_0) \approx q(\mathbf{z}_T)$. It then learns a reverse process $p_\theta(\mathbf{z}_{0:T})$, so that the generative process consists in sequentially denoising a random noise distributed according to $q(\mathbf{z}_T)$.

The standard Gaussian DDPM (Ho et al., 2020) defines the forward process as a Markov chain $q_{\mathrm{M}}(\mathbf{z}_{1:T} \,|\, \mathbf{z}_0)$, where the subscript $\mathrm{M}$ means "Markovian". At each time step $t$, Gaussian noise is added according to a predefined variance schedule $(\beta_t)_{0 \leq t \leq T}$, with a joint density given

by $q_{\mathrm{M}}(\mathbf{z}_{0:T}) = q(\mathbf{z}_0) \prod_{t=1}^{T} q_{\mathrm{M}}(\mathbf{z}_t \,|\, \mathbf{z}_{t-1})$, and transitions $q_{\mathrm{M}}(\mathbf{z}_t \,|\, \mathbf{z}_{t-1}) = \mathcal{N}(\mathbf{z}_t; \sqrt{1-\beta_t}\mathbf{z}_{t-1}, \beta_t \mathbf{I})$. The variance schedule is chosen so that $q_{\mathrm{M}}(\mathbf{z}_T)$ is very close to a standard isotropic Gaussian distribution, i.e., $q_{\mathrm{M}}(\mathbf{z}_T) \approx \mathcal{N}(\mathbf{z}_t; 0, \mathbf{I})$.

The reverse process $p_\theta(\mathbf{z}_{0:T})$ is the generative model. The optimal parameters are found by maximizing the Evidence Lower-Bound (ELBO), whose expression is given by $\mathrm{ELBO}_{\mathrm{M}}(\theta) = \mathbb{E}[\log p_\theta(\mathbf{z}_0 \,|\, \mathbf{z}_1) - \sum_{t=2}^{T} L_t(\theta)]$, where we let $L_t(\theta) = D_{\mathrm{KL}}(q_{\mathrm{M}}(\mathbf{z}_{t-1} \,|\, \mathbf{z}_t, \mathbf{z}_0) \,\|\, p_\theta(\mathbf{z}_{t-1} \,|\, \mathbf{z}_t))$. The formulation of the DDPM forward process has two major consequences. First, the true reverse model has a Markovian structure, allowing for a sequential generation algorithm. Second, in the expression of $\mathcal{L}_{\mathrm{M}}$, the posterior density $q_{\mathrm{M}}(\mathbf{z}_{t-1} \,|\, \mathbf{z}_t, \mathbf{z}_0)$ is Gaussian. Straightforward computations show that, letting $\bar{\alpha}_t = \prod_{s=1}^{t}(1-\beta_s)$, the mean and covariance are given by

$$\tilde{\mu}_t(\mathbf{z}_t, \mathbf{z}_0) = \frac{\sqrt{1-\beta_t}(1-\bar{\alpha}_{t-1})}{1-\bar{\alpha}_t}\mathbf{z}_t + \frac{\sqrt{\bar{\alpha}_{t-1}}\beta_t}{1-\bar{\alpha}_t}\mathbf{z}_0, \quad (1)$$

$$\text{and} \quad \tilde{\beta}_t = \frac{1-\bar{\alpha}_{t-1}}{1-\bar{\alpha}_t}\beta_t. \quad (2)$$

This property is a direct consequence of the choices that define the model, particularly the Markovian structure and Gaussianity. Extending this framework to other distributions is challenging because it requires designing processes where this posterior remains computable.

### 3.2. A Non-Markovian Framework for Diffusion Models

To overcome the limited flexibility of the standard DDPM, one can define a non-Markovian forward process $q_{\mathrm{NM}}(\mathbf{z}_{0:T})$. Since this is the setting we consider in the remainder of the paper, we drop the subscript NM for notational ease. Specifically, we consider a star-shaped forward process (Okhotin et al., 2023).

**Assumption 3.1.** We suppose that the forward process $q(\mathbf{z}_{0:T})$ has the following star-shaped form:

$$q(\mathbf{z}_{0:T}) = q(\mathbf{z}_0) \prod_{t=1}^{T} q(\mathbf{z}_t \,|\, \mathbf{z}_0). \quad (3)$$

Note that all the latents $(\mathbf{z}_t)_{1 \leq t \leq T}$ are conditionally independent given $\mathbf{z}_0$. This contrasts with DDPMs, where the forward process generates each $\mathbf{z}_t$ from $\mathbf{z}_{t-1}$, and the Markov property implies that $\mathbf{z}_t$ contains all the necessary information from the trajectory $\mathbf{z}_{t:T}$ to predict $\mathbf{z}_{t-1}$. In this non-Markovian setting, $\mathbf{z}_{(t+1):T}$ carries information about $\mathbf{z}_0$ that is not contained in $\mathbf{z}_t$ alone. The true reverse process is non-Markovian, $q(\mathbf{z}_{0:T}) = q(\mathbf{z}_T) \prod_{t=1}^{T} q(\mathbf{z}_{t-1} \,|\, \mathbf{z}_{t:T})$. Thus, we approximate it by a parametric model $p_\theta(\mathbf{z}_{0:T}) = p_\theta(\mathbf{z}_T) \prod_{t=1}^{T} p_\theta(\mathbf{z}_{t-1} \,|\, \mathbf{z}_{t:T})$ that conditions on the whole tail $\mathbf{z}_{t:T}$. The corresponding ELBO writes $\mathrm{ELBO}(\theta) = \mathbb{E}[\log p_\theta(\mathbf{z}_0 \,|\, \mathbf{z}_{1:T}) -$

$\sum_{t=2}^{T} D_{\mathrm{KL}}(q(\mathbf{z}_{t-1} \,|\, \mathbf{z}_0) \,\|\, p_\theta(\mathbf{z}_{t-1} \,|\, \mathbf{z}_{t:T}))]$. Note that only the marginals $q(\mathbf{z}_{t-1} \,|\, \mathbf{z}_0)$ are needed to compute it, allowing the use of various noising distributions.

### 3.3. Efficient Reversal via a Sufficient Tail Statistic

The main challenge in this framework is the conditioning on the entire future trajectory $p_\theta(\mathbf{z}_{t-1} \,|\, \mathbf{z}_{t:T})$, which becomes computationally impractical as the sequence length increases. A solution is to compress the tail $\mathbf{z}_{t:T}$ into a compact, fixed-size form, without losing relevant information about the reverse step (Okhotin et al., 2023). Formally, this means defining a sufficient tail statistic $\mathsf{s}_t$ such that $q(\mathbf{z}_{t-1} \,|\, \mathbf{z}_{t:T}) = q(\mathbf{z}_{t-1} \,|\, \mathsf{s}_t)$. We require two assumptions about the forward process $q(\mathbf{z}_t \,|\, \mathbf{z}_0)$ to achieve this.

**Assumption 3.2 (Exponential Family Latent Transitions).** The conditional distribution $q(\mathbf{z}_t \,|\, \mathbf{z}_0)$ on $\mathbb{R}^d$ is an exponential family density with respect to a reference measure $\nu$,

$$q(\mathbf{z}_t \,|\, \mathbf{z}_0) = h_t(\mathbf{z}_t) \exp\Big[\langle \eta_t(\mathbf{z}_0), \psi_t(\mathbf{z}_t) \rangle - \mathsf{a}_t(\mathbf{z}_0)\Big], \quad (4)$$

where $\psi : \mathbb{R}^d \to \mathbb{R}^m$ is a sufficient statistic of the distribution, $\eta_t(\mathbf{z}_0) \in \mathbb{R}^m$ is the natural parameter, $\langle \cdot, \cdot \rangle$ denotes the canonical inner-product on $\mathbb{R}^m$, $h_t : \mathbb{R}^d \to \mathbb{R}$, and $\mathsf{a}_t(\mathbf{z}_0)$ is the log-partition function.

**Assumption 3.3 (Linear parameterization).** The natural parameter $\eta_t(\mathbf{z}_0) \in \mathbb{R}^m$ in (4) depends linearly on a feature map $f : \mathbb{R}^d \to \mathbb{R}^n$ of the initial state $\mathbf{z}_0$:

$$\eta_t(\mathbf{z}_0) = \mathbf{A}_t f(\mathbf{z}_0) + \mathbf{b}_t, \quad (5)$$

where the matrix $\mathbf{A}_t \in \mathbb{R}^{m \times n}$ and the bias vector $\mathbf{b}_t \in \mathbb{R}^m$ are time-dependent parameters. Moreover, (4) defines a regular minimal exponential family.

Under these assumptions, the following theorem (Okhotin et al., 2023) constructs a valid sufficient tail statistic. For completeness, we provide a short proof in Appendix A.1.

**Theorem 3.4.** *Under Assumptions 3.2 and 3.3, the reverse conditional distribution $q(\mathbf{z}_{t-1} \,|\, \mathbf{z}_{t:T})$ depends on the tail sequence $\mathbf{z}_{t:T}$ only through the sufficient tail statistic $\mathsf{s}_t$ defined as*

$$\mathsf{s}_t = \sum_{k=t}^{T} \mathbf{A}_k^\top \psi(\mathbf{z}_k). \quad (6)$$

*That is, $q(\mathbf{z}_{t-1} \,|\, \mathbf{z}_{t:T}) = q(\mathbf{z}_{t-1} \,|\, \mathsf{s}_t)$.*

In the original SS-DDPM formulation, the approximate reverse step is $p_\theta(\mathbf{z}_{t-1} \,|\, \mathbf{z}_{t:T}) = q(\mathbf{z}_{t-1} \,|\, \mathbf{z}_0 = \mathbf{z}_\theta(\mathsf{s}_t, t))$. Hence, the model does not condition on $\mathbf{z}_t$ to predict $\mathbf{z}_{t-1}$. Instead, it relies solely on $\mathsf{s}_t$. While the sufficient statistics form a Markov chain, the temporal coherence between timesteps can be compromised if, for instance, the prediction of $\mathbf{z}_0$ fluctuates between steps. This can result in degraded

sampling capabilities. In the next section, we propose a model that utilizes information from $\mathbf{z}_t$ to construct $\mathbf{z}_{t-1}$, while retaining the benefits of SS-DDPM.

**Why temporal incoherence hurts few-step generation.** The SS-DDPM reverse process would be viable with a perfect oracle and poses no issue at a distributional level, but it does not prevent jumps across the data manifold, and high variance in the predicted $\hat{\mathbf{z}}_0$ cripples the model's generative capabilities in practice. Intuitively, SS-DDPM requires many timesteps for $\hat{\mathbf{z}}_0$ fluctuations to average out; temporal incoherence therefore primarily affects few-step generation. We make this argument quantitative in Appendix E.4. We find that M-Star reduces $\mathrm{Var}(\hat{\mathbf{z}}_0)$ by $\sim 53\%$ relative to SS-DDPM on both ImageNet-64 and ImageNet-128.

## 4. Markovian-Projected Star-Shaped Diffusion

This section formalizes Markovian-Projected Star-Shaped Diffusion (M-Star). We define the model and explain design choices. The training and sampling procedures are respectively described in Algorithm 1 and Algorithm 2.

### 4.1. Model definition

M-Star is a hybrid approach that assumes the star-shaped forward process of Assumption 3.1 and exponential family distributions as in Assumptions 3.2 and 3.3. However, instead of relying on the conditional independence property, we introduce a dual Markovian process that serves as a bridge for the reverse sampling steps. Gyöngy's mimicking theorem (Gyöngy, 1986), and more generally *Markovian projection* in mathematical finance, offer a useful conceptual anchor. Gyöngy's result states that a continuous-time non-Markovian Itô process admits a Markov process with the same one-dimensional marginals. We invoke it purely as motivational context; M-Star is developed in discrete time and the construction below is self-contained, directly solving the Chapman–Kolmogorov equation rather than invoking the continuous-time result.

**Markovian Projection.** Fix $\mathbf{z}_0$ and let $\mu_t$ denote the marginal distribution $q(\mathbf{z}_t \mid \mathbf{z}_0)$ implied by the star-shaped forward process of Assumption 3.1. We seek to define a Markov process $(\tilde{\mathbf{z}}_t)_{0 \le t \le T}$ characterized by transition kernels $K_t(\,\cdot\,,\,\cdot \mid \mathbf{z}_0) : \mathbb{R}^d \times \mathcal{B}(\mathbb{R}^d) \to [0,1]$ such that for all $t \in [\![1, T]\!]$, the marginal distribution of $\tilde{\mathbf{z}}_t$ is equal to $\mu_t$. That is, for any measurable set $A \in \mathcal{B}(\mathbb{R}^d)$,

$$\mathbb{P}(\tilde{\mathbf{z}}_t \in A \mid \tilde{\mathbf{z}}_{t-1}, \mathbf{z}_0) = K_t(\tilde{\mathbf{z}}_{t-1}, A \mid \mathbf{z}_0),$$

$$\text{and} \quad \mathbb{P}(\tilde{\mathbf{z}}_t \in A \mid \mathbf{z}_0) = \mu_t(A) = \int_A q(\mathbf{z}_t \mid \mathbf{z}_0)\,\nu(\mathrm{d}\mathbf{z}_t).$$

Assume that the kernel $K_t$ admits a density with respect to the reference measure $\nu$, denoted by $k_t(\tilde{\mathbf{z}}_t \mid \mathbf{z}_{t-1}, \mathbf{z}_0)$:

$$K_t(\mathbf{z}_{t-1}, \mathrm{d}\tilde{\mathbf{z}}_t \mid \mathbf{z}_0) = k_t(\tilde{\mathbf{z}}_t \mid \mathbf{z}_{t-1}, \mathbf{z}_0)\,\nu(\mathrm{d}\tilde{\mathbf{z}}_t).$$

The density $k_t$ ought to be a the solution to the Chapman-Kolmogorov integral equation:

$$q(\mathbf{z}_t \mid \mathbf{z}_0) = \int k_t(\mathbf{z}_t \mid \mathbf{z}_{t-1}, \mathbf{z}_0) q(\mathbf{z}_{t-1} \mid \mathbf{z}_0)\nu(\mathrm{d}\mathbf{z}_{t-1}). \quad (7)$$

Now consider the Markovian reverse process. Let $\tilde{q}$ denote the densities of the mimicking process. Bayes' rule gives

$$\tilde{q}(\mathbf{z}_{t-1} \mid \mathbf{z}_t, \mathbf{z}_0) = \frac{\tilde{q}(\mathbf{z}_t \mid \mathbf{z}_{t-1}, \mathbf{z}_0)\,\tilde{q}(\mathbf{z}_{t-1} \mid \mathbf{z}_0)}{\tilde{q}(\mathbf{z}_t \mid \mathbf{z}_0)}.$$

By definition of $k_t$ and construction of the reversal, we get

$$\tilde{q}(\mathbf{z}_{t-1} \mid \mathbf{z}_t, \mathbf{z}_0) \propto k_t(\mathbf{z}_t \mid \mathbf{z}_{t-1}, \mathbf{z}_0)\,q(\mathbf{z}_{t-1} \mid \mathbf{z}_0). \quad (8)$$

The factor $q(\mathbf{z}_{t-1} \mid \mathbf{z}_0)$ suggests an exponential family form for the mimicking posterior, but this is only achievable if the kernel $k_t$ can be defined to be compatible with this structure.

**Kernel derivation.** We now establish a valid kernel density $k_t$. It is clear that if $\pi_t(\mathbf{z}_{t-1} \mid \mathbf{z}_t, \mathbf{z}_0)$ is any valid auxiliary density, meaning that its integral equals 1, defining

$$k_t(\mathbf{z}_t \mid \mathbf{z}_{t-1}, \mathbf{z}_0) = q(\mathbf{z}_t \mid \mathbf{z}_0)\frac{\pi_t(\mathbf{z}_{t-1} \mid \mathbf{z}_t, \mathbf{z}_0)}{q(\mathbf{z}_{t-1} \mid \mathbf{z}_0)} \quad (9)$$

guarantees that $k_t$ satisfies (7); see Appendix B.1.

To find a suitable auxiliary density, recall that we work under Assumptions 3.2 and 3.3. Therefore, we specify the density $\pi_t(\mathbf{z}_{t-1} \mid \mathbf{z}_t, \mathbf{z}_0)$ so that it lies in the conjugate exponential family. Let $\tilde{\eta}_{t-1}(\mathbf{z}_0, \mathbf{z}_t) = \eta_{t-1}(\mathbf{z}_0) + \mathbf{M}_t\psi(\mathbf{z}_t)$, where $\mathbf{M}_t$ is an interaction matrix. We define

$$\pi_t(\mathbf{z}_{t-1} \mid \mathbf{z}_t, \mathbf{z}_0) = h_{t-1}(\mathbf{z}_{t-1})$$
$$\times \exp\Big[\langle\tilde{\eta}_{t-1}(\mathbf{z}_0, \mathbf{z}_t), \psi(\mathbf{z}_{t-1})\rangle - \tilde{\mathsf{a}}_{t-1}(\mathbf{z}_0, \mathbf{z}_t)\Big], \quad (10)$$

where $\tilde{\mathsf{a}}_{t-1}(\mathbf{z}_0, \mathbf{z}_t)$ ensures that $\pi$ integrates to 1.

This directly specifies the posterior $\tilde{q}(\mathbf{z}_{t-1} \mid \mathbf{z}_t, \mathbf{z}_0)$ in (8) as an exponential family density, with natural parameter

$$\tilde{\eta}_{t-1}(\mathbf{z}_0, \mathbf{z}_t) = \eta_{t-1}(\mathbf{z}_0) + \mathbf{M}_t\psi(\mathbf{z}_t). \quad (11)$$

**Markovian transitions, non-Markovian guidance.** M-Star is a hybrid framework: the transition from $\mathbf{z}_t$ to $\mathbf{z}_{t-1}$ is Markovian, but the $\hat{\mathbf{z}}_0$ that guides each noisy state integrates non-Markovian memory through the tail statistic. Formally, the forward joint distribution of the projected process factorizes as $\tilde{q}(\mathbf{z}_{0:T}) = q(\mathbf{z}_0)\prod_{t=1}^{T} k_t(\mathbf{z}_t \mid \mathbf{z}_{t-1}, \mathbf{z}_0)$. The true unconditional reversal is also Markovian, $\tilde{q}(\mathbf{z}_{t-1} \mid \mathbf{z}_t) = \int \tilde{q}(\mathbf{z}_{t-1} \mid \mathbf{z}_t, \mathbf{z}_0)\,\tilde{q}(\mathbf{z}_0 \mid \mathbf{z}_t)\,\nu(\mathrm{d}\mathbf{z}_0)$. In practice, the integral is approximated by a point estimate $\hat{\mathbf{z}}_0 = \mathbf{z}_\theta(\mathsf{s}_t, t)$. Since this guidance signal relies on the full tail statistic $\mathsf{s}_t$, the resulting sampler is non-Markovian: M-Star pairs a Markovian local transition (the source of stability and temporal coherence) with a non-Markovian global memory (the source of expressiveness across exponential families).

**Algorithms.** The training and sampling loops for M-Star are given in Algorithms 1 and 2.

---

**Algorithm 1** M-Star Training Procedure

---

**repeat**

    $\mathbf{z}_0 \sim q(\mathbf{z}_0)$

    $t \sim \text{Uniform}(\{1, \ldots, T\})$

    Sample $(\mathbf{z}_k)_{k=t}^T$ where $\mathbf{z}_k \sim q(\mathbf{z}_k \,|\, \mathbf{z}_0)$

    Compute $\mathsf{s}_t = \sum_{k=t}^T \mathbf{A}_k^\top \psi(\mathbf{z}_k)$

    Predict $\hat{\mathbf{z}}_0 = \mathbf{z}_\theta(\mathsf{s}_t, t)$

    Take a gradient step on $\nabla_\theta \mathcal{L}(\theta)$

                        $\{$Typically minimizing $\|\mathbf{z}_0 - \hat{\mathbf{z}}_0\|^2\}$

    Update $\mathbf{M}_t$ to minimize (12)

**until** converged

**return** $(\theta, (\mathbf{M}_t)_{1 \le t \le T})$

---

**Algorithm 2** M-Star Sampling Procedure

---

Sample $\mathbf{z}_T \sim q(\mathbf{z}_T)$

$\mathsf{s}_T \leftarrow \mathbf{A}_T^\top \psi(\mathbf{z}_T)$

**for** $t = T$ **down to** $1$ **do**

    $\hat{\mathbf{z}}_0 \leftarrow \mathbf{z}_\theta(\mathsf{s}_t, t)$

    Sample $\mathbf{z}_{t-1} \sim \tilde{q}(\mathbf{z}_{t-1} \,|\, \mathbf{z}_t, \mathbf{z}_0 = \tilde{\mathbf{z}}_0)$ from (11)

    **if** $t > 1$ **then**

        $\mathsf{s}_{t-1} \leftarrow \mathsf{s}_t + \mathbf{A}_{t-1}^\top \psi(\mathbf{z}_{t-1})$

    **end if**

**end for**

**return** $\hat{\mathbf{z}}_0$

---

## 4.2. Interaction Matrix

The interaction matrix $\mathbf{M}_t$, which governs the reverse kernel through $\tilde{q}(\mathbf{z}_{t-1} \,|\, \mathbf{z}_t, \mathbf{z}_0)$. Formally, any choice of $\mathbf{M}_t$ yields a valid projected forward process, as per the previous section and Appendix B.1. However, tuning $\mathbf{M}_t$ is critical to the effectiveness of the method. If $\mathbf{M}_t = 0$, the process ignores $\mathbf{z}_{t-1}$ and reverts to SS-DDPM. If $\mathbf{M}_t$ is too large, the process ignores the global guidance from $\mathbf{z}_0$. **Loss function.** It is natural to ask whether, among all the forward processes that stem from the definition in (10), there exists one that is optimal, in a sense to be defined. We propose to learn the matrix $\mathbf{M}_t$ by taking gradient steps on

$$\mathcal{L}(\mathbf{M}_t) = D_{\text{KL}}(\tilde{q}(\mathbf{z}_{t-1} \,|\, \mathbf{z}_t, \mathbf{z}_0) \,\|\, \tilde{q}(\mathbf{z}_{t-1} \,|\, \mathbf{z}_t, \mathbf{z}_0 = \hat{\mathbf{z}}_0)), \quad (12)$$

where $\tilde{q}(\mathbf{z}_{t-1} \,|\, \mathbf{z}_t, \mathbf{z}_0)$ is defined via (11), and $\hat{\mathbf{z}}_0 = \mathbf{z}_\theta(\mathsf{s}_t, t)$ is our prediction. In this expression, the target density $\tilde{q}(\mathbf{z}_{t-1} \,|\, \mathbf{z}_t, \mathbf{z}_0)$ is fixed, and the learning of $\mathbf{M}_t$ only affects $\tilde{q}(\mathbf{z}_{t-1} \,|\, \mathbf{z}_t, \mathbf{z}_0 = \hat{\mathbf{z}}_0)$. Specifically, let $\mathbf{M}_t^\circ$ be a copy of the interaction matrix detached from the computational graph. Assuming $\eta^\circ = \eta_{t-1}(\mathbf{z}_0) + \mathbf{M}_t^\circ \psi(\mathbf{z}_t)$ and $\hat{\eta} = \eta_{t-1}(\hat{\mathbf{z}}_0) + \mathbf{M}_t \psi(\mathbf{z}_t)$ denote the natural parameters for each density, from Appendix B.2 the divergence in (12) is

$$\mathcal{L}(\mathbf{M}_t) = \tilde{\mathsf{a}}(\hat{\eta}) - \tilde{\mathsf{a}}(\eta^\circ) - \langle \nabla \tilde{\mathsf{a}}(\eta^\circ), \hat{\eta} - \eta^\circ \rangle,$$

where by abuse of notation we parameterize the log-partition $\tilde{\mathsf{a}}$ by the natural parameter. The gradient of this loss is

$$\nabla \mathcal{L}(\mathbf{M}_t) = (\nabla \tilde{\mathsf{a}}(\hat{\eta}) - \nabla \tilde{\mathsf{a}}(\eta^\circ)) \, \psi(\mathbf{z}_t)^\top$$
$$= (\mathbb{E}_{\hat{\eta}}[\psi] - \mathbb{E}_{\eta^\circ}[\psi]) \, \psi(\mathbf{z}_t)^\top.$$

**Intuition.** The loss $\mathcal{L}(\mathbf{M}_t)$ measures the discrepancy between the ideal next step (knowing $\mathbf{z}_0$) and the model's actual step (using $\hat{\mathbf{z}}_0$). Note that a perfect $\mathbf{z}_0$-predictor leads to a null gradient. It quantifies how strongly we should anchor the next state to the current one so the trajectory remains robust to prediction errors. Over the entire dataset, minimizing this objective amounts to solving the normal equations of an ordinary least squares regression. The projection along $\psi(\mathbf{z}_t)$ means that the interaction matrix only compensates for the part of the error that correlates with the current noisy state. The trajectory is pulled back to $\mathbf{z}_t$ in the directions where the global prediction $\hat{\mathbf{z}}_0$ is less reliable, compensating for the U-Net's biases at the dataset level. Early in training, $\mathbf{M}_t$ aggressively compensates for large reconstruction errors; as training converges, it settles to a value reflecting the noise schedule structure (Figures 3 and 7 to 9). While we could learn state-dependent interaction matrices $\mathbf{M}_t(\mathbf{z}_t)$, this amortized approach is a sensible tradeoff between the expressiveness of the error corrector and the computational overhead.

**Decomposition in state coordinates.** Substituting Assumption 3.3 into (11) makes the natural parameter transparent in state coordinates:

$$\tilde{\eta}_{t-1}(\mathbf{z}_0, \mathbf{z}_t) = \underbrace{\mathbf{A}_{t-1}\, f(\mathbf{z}_0)}_{\text{global hub}} + \underbrace{\mathbf{b}_{t-1}}_{\text{schedule bias}} + \underbrace{\mathbf{M}_t\, \psi(\mathbf{z}_t)}_{\text{local Markov}}. \quad (13)$$

The three terms admit a clean interpretation: the first carries global guidance from the clean data hub, the second encodes the noise-schedule bias, and the third is the local coupling that anchors the next state to the current one. When $\mathbf{M}_t$ is the identity (Gaussian case), this expression lives directly in state space and recovers exactly the DDPM posterior mean (Section 4.4 and Appendix B.3). In the general case, the map from natural to mean parameters goes through $\nabla \tilde{\mathsf{a}}$, which is distribution-specific, and a deterministic sampler is generally unavailable under arbitrary exponential-family noise. This decomposition also clarifies how M-Star relates to existing reparameterizations: DDIM (Song et al., 2020a) keeps the global term and removes the local stochasticity, while ReMDM (Lou et al., 2023) plays an analogous role in discrete state spaces. M-Star instead constructs a Markovian projection by working directly in the natural parameter space.

### 4.3. Noise Schedules

Completing the model definition requires specifying the trajectories of the parameters $(\mathbf{A}_t)$ and $(\mathbf{b}_t)$ from Assump-

tion 3.3, as well as the interaction matrices ($\mathbf{M}_t$). All of these choices will come as consequences of the noise schedule design. However, defining a noise schedule directly on the parameters of a non-standard distribution can be difficult. We circumvent this issue by relying on the asymptotic behavior of the sufficient statistics.

**Asymptotic Normality of the Tail Statistic.** To justify using robust noise schedules from the Gaussian DDPM literature, we leverage the Central Limit Theorem (CLT). Since the tail statistic $\mathsf{s}_t$ is a sum of independent random vectors conditionally to $\mathbf{z}_0$, its distribution converges to a multivariate Gaussian as the tail length grows. Hence, regardless of the exponential family being used, the effective noise corrupting the signal in the sufficient statistic domain becomes Gaussian noise. This justifies transferring noise schedules from the Gaussian DDPM literature because the noise profiles are expected to turn out very similar.

**Theorem 4.1.** *Let $q(\mathbf{z}_{0:T})$ be the star-shaped forward process defined under Assumptions 3.1 to 3.3, and let $\mathsf{s}_t = \sum_{k=t}^{T} \mathbf{A}_k^\top \psi(\mathbf{z}_k)$ be the sufficient tail statistic. Define the conditional mean $\mu_N(\mathbf{z}_0) = \mathbb{E}[\mathsf{s}_t \mid \mathbf{z}_0]$ and covariance $B_N(\mathbf{z}_0) = \mathrm{Cov}(\mathsf{s}_t \mid \mathbf{z}_0)$, where $N = T - t + 1$ denotes the tail length. Under Assumptions A.1 and A.2, as $N \to \infty$,*

$$B_N(\mathbf{z}_0)^{-1/2}(\mathsf{s}_t - \mu_N(\mathbf{z}_0)) \xrightarrow{(d)} \mathcal{N}(0, \mathbf{I}),$$

*where $\xrightarrow{(d)}$ denotes the convergence in distribution. Consequently, the conditional density $q(\mathsf{s}_t \mid \mathbf{z}_0)$ approximates a heteroscedastic Gaussian density $\mathcal{N}(\mathsf{s}_t; \mu_N(\mathbf{z}_0), B_N(\mathbf{z}_0))$.*

The proof (see Appendix A.2) relies on a generalized CLT (Billingsley, 2013), extended to the multidimensional case using the Cramér-Wold theorem (Cramér & Wold, 1936).

Theorem 4.1 shows that the tail statistic asymptotically resembles a Gaussian distribution with mean $\mu_N(\mathbf{z}_0)$ and covariance matrix $B_N(\mathbf{z}_0)$. Thus, it makes sense to compare the signal-to-noise ratio (SNR) of $\mathsf{s}_t$, defined by:

$$\mathrm{SNR}(\mathsf{s}_t) = \frac{\|\mathbb{E}[\mathsf{s}_t \mid \mathbf{z}_0]\|^2}{\mathrm{Tr}(\mathrm{Cov}(\mathsf{s}_t \mid \mathbf{z}_0))} = \frac{\|\mu_N(\mathbf{z}_0)\|^2}{\mathrm{Tr}(B_N(\mathbf{z}_0))},$$

to the the SNR defined by the noise schedule in a Gaussian DDPM. This SNR is determined by $(\bar{\alpha}_t)_{0 \leq t \leq T}$, through the formula $\mathrm{SNR}_{\mathrm{DDPM}}(t) = \bar{\alpha}_t/(1 - \bar{\alpha}_t)$. Thus, to transfer schedules from the Gaussian DDPM literature to the M-Star framework, we simply match the SNRs.

The theorem holds as $N = T - t + 1 \to \infty$. This means the Gaussian approximation is valid near the data (i.e., when $t \to 0$, with $N \gg 1$), which is encouraging because this is where precise gradients matter the most for sample quality. On the other hand, the approximation is weakest near the noise ($t \to T$, $N \to 1$). It would seem the signal is weak enough in this regime that the resulting errors have

negligible impact on the final sample quality. Moreover, this regime is mostly skipped if we distill the model (see Section 4.5).

**Boundary Conditions.** The forward process should be constructed such that when $t \to 0$, the conditional distribution $q(\mathbf{z}_t \mid \mathbf{z}_0)$ converges to a Dirac delta centered at the true data, denoted by $\delta_{\mathbf{z}_0}$. In contrast, as $t$ moves towards $T$, $q(\mathbf{z}_t \mid \mathbf{z}_0)$ should gradually degrade into a distribution that is independent of the initial state $\mathbf{z}_0$. Formally, we require

$$q(\mathbf{z}_t \mid \mathbf{z}_0) \xrightarrow[t \to 0]{} \delta_{\mathbf{z}_0}(\mathbf{z}_t), \quad \text{and} \quad q(\mathbf{z}_t \mid \mathbf{z}_0) \xrightarrow[t \to T]{} q(\mathbf{z}_T).$$

For exponential family distributions that satisfy Assumption 3.3, these conditions can be met by carefully designing the time-dependent matrices ($\mathbf{A}_t$) and bias vectors ($\mathbf{b}_t$) as functions of the predefined noise schedule. While these requirements may appear abstract, the construction of such parameterizations is actually intuitive in most cases. We provide a concrete example with the von Mises-Fisher distribution in Appendix C.

### 4.4. Equivalence with DDPMs in the Gaussian case

We now demonstrate that M-Star recovers the standard Gaussian DDPM (Ho et al., 2020) as a special case, reinforcing the theoretical grounding of our method. We use the same notation as in Section 3.1 for the DDPM noise schedule.

**Posterior Matching.** In Assumption 3.1, suppose we take $q(\mathbf{z}_t \mid \mathbf{z}_0) = \mathcal{N}(\mathbf{z}_t; \sqrt{\bar{\alpha}_t}\mathbf{z}_0, (1 - \bar{\alpha}_t)\mathbf{I})$. Identifying $\psi(\mathbf{z}) = \mathbf{z}$, and $\eta_t(\mathbf{z}_0) = \sqrt{\bar{\alpha}_t}/(1 - \bar{\alpha}_t)\mathbf{z}_0$, this satisfies Assumption 3.2. Moreover, Assumption 3.3 is verified with $\mathbf{A}_t = \sqrt{\bar{\alpha}_t}/(1 - \bar{\alpha}_t)$, $f(\mathbf{z}) = \mathbf{z}$, and $\mathbf{b}_t = 0$. We define the interaction matrix $\mathbf{M}_t$ as

$$\mathbf{M}_t = \frac{\sqrt{\alpha_t}}{\beta_t} \mathbf{I}. \tag{14}$$

The M-Star reverse posterior $\tilde{q}(\mathbf{z}_{t-1} \mid \mathbf{z}_t, \mathbf{z}_0)$ is a Gaussian density whose natural parameter is given by (11). Substituting the specified $\eta_{t-1}(\mathbf{z}_0)$ and $\mathbf{M}_t$ into (11), we obtain

$$\tilde{\eta}_{t-1}(\mathbf{z}_0, \mathbf{z}_t) = \eta_{t-1}(\mathbf{z}_0) + \mathbf{M}_t \mathbf{z}_t = \frac{\sqrt{\bar{\alpha}_{t-1}}}{1 - \bar{\alpha}_{t-1}} \mathbf{z}_0 + \frac{\sqrt{\alpha_t}}{\beta_t} \mathbf{z}_t.$$

We compare the density with the DDPM posterior given in Section 3.1. For a Gaussian distribution with mean $\mu$ and covariance matrix $\Sigma$, the natural parameter is $\eta = \Sigma^{-1}\mu$. With $\mu = \tilde{\mu}_t(\mathbf{z}_t, \mathbf{z}_0)$ and $\Sigma = \tilde{\beta}_t \mathbf{I}$ from (1) and (2), we get:

$$\tilde{\beta}_t^{-1} \tilde{\mu}_t(\mathbf{z}_t, \mathbf{z}_0) = \tilde{\eta}_{t-1}(\mathbf{z}_0, \mathbf{z}_t).$$

Detailed derivations are given in Appendix B.3. This implies that under particular design choices, the M-Star sampling process recovers the Gaussian DDPM reverse process.

**Learnt Interaction Matrix.** While we imposed an analytical form of $\mathbf{M}_t$ in the derivations above, in practice

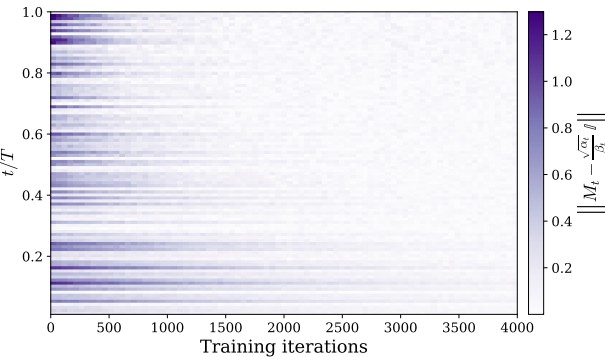

*Figure 3.* Heatmap of the convergence error $\|\mathbf{M}_t - \frac{\sqrt{\alpha_t}}{\beta_t}\mathbf{I}\|$ over training iterations and diffusion timesteps $t/T$ on CIFAR-100. Higher residual error near $t/T \to 1$ reflects the increased optimization difficulty at the noisy end of the trajectory rather than a failure of the CLT approximation: the regime $N \to \infty$ in which Theorem 4.1 is most accurate corresponds to $t \to 0$.

we treat it as a learnable parameter optimizing (12). Strikingly, empirical results show that the learnt $\mathbf{M}_t$ converges to the theoretical value in (14). This is visualized in Figure 3, confirming that M-Star naturally recovers standard DDPM dynamics from scratch when assuming a Gaussian star-shaped process, providing additional validation for our framework.

### 4.5. Accelerated Sampling via Progressive Distillation

While the Markovian projection introduced in Section 4 ensures a stable and coherent generative trajectory, the requirement of sequential denoising steps remains a significant computational bottleneck. To address this, we propose a distillation framework that leverages the unique structural properties of the M-Star framework. Unlike standard diffusion models that rely on local score or noise-error matching, M-Star is inherently structured as a hub-and-spoke model, which we argue is more amenable to the large-step mappings required by distillation (Salimans & Ho, 2022).

The fundamental strength of M-Star in the context of distillation lies in its training objective. A standard DDPM teacher typically learns to predict the infinitesimal noise component, making multi-step leaps difficult to approximate due to high discretization error. In contrast, our teacher model is explicitly trained to reconstruct the clean data hub from the sufficient tail statistic. This global awareness provides a more stable and informative target for a student model. Specifically, since the teacher already possesses the ability to estimate the final manifold from any intermediate state, the student's task is reduced to learning a consistency mapping between compressed statistics of the noise tail.

**Implementation from Pre-trained Teacher.** We assume access to a fully converged teacher model trained with $T$ steps. We initialize a student model with the teacher's weights and iteratively halve the number of sampling steps.

For a given student step, the procedure is as follows:

1. **Teacher Trajectory:** Given a noisy state $\mathbf{z}_t$ and its corresponding tail statistic $\mathbf{s}_t$, the teacher computes the denoised estimate. Using (11), we perform two sequential denoising steps to reach $\mathbf{z}_{t-2}$, yielding an updated tail statistic $\mathbf{s}_{t-2}$.

2. **Consistency Target:** The student is trained to directly predict the teacher's subsequent estimate using only the initial statistic. We minimize the weighted mean-squared error between the student's prediction and the teacher's "two-step" lookahead:

$$\mathcal{L}_{\text{distill}}(\phi) = \mathbb{E}_t\left[\mathbb{E}_{\mathbf{s}_t}\left[\|\mathbf{z}_\phi(\mathbf{s}_t, t) - \mathbf{z}_\theta(\mathbf{s}_{t-1}, t-1)\|^2\right]\right].$$

**Trajectory Straightening and Information Density.** We posit that M-Star's superior performance at low step counts (see Figure 1) is due to two factors. First, the Markovian projection effectively "straightens" the sampling trajectory compared to the independent spokes of SS-DDPM, reducing the variance the student must model. Second, because $\mathbf{s}_t$ is a sufficient statistic for the entire noise tail, the student receives a more feature-rich representation of the remaining noise than standard models that only observe the current state. This enables high-fidelity one-step generation that remains anchored to the data manifold, a property particularly vital for the heavy-tailed and constrained distributions found in meteorological data.

## 5. Experiments

We validate our method through experiments on standard image benchmarks including MNIST, CIFAR, and ImageNet. We then evaluate M-Star in a class-conditional image generation setting on ImageNet (Deng et al., 2009), a commonly used reference for large-scale generative modeling. To better isolate the effects of key design decisions, we additionally carry out a set of ablation studies on ImageNet, examining in particular the influence of classifier-free guidance and other important hyperparameter choices. Further details are provided in Appendix E.

### 5.1. Results on CIFAR and MNIST

We evaluate M-Star on unsupervised anomaly detection and generative modeling tasks. Table 1 reports AUPRC scores for unsupervised anomaly detection across MNIST and CIFAR datasets. M-Star consistently outperforms baseline methods, including VAE-based, GAN-based, and energy-based models.

### 5.2. Experimental Results on ImageNet

We benchmark M-Star against state-of-the-art distillation methods using Fréchet Inception Distance (FID) for fidelity

*Table 1.* AUPRC ($\mathcal{O}$) scores for unsupervised anomaly detection on MNIST and CIFAR. Results averaged over 10 trials; $\pm$ signs indicate standard deviations.

| Model | MNIST Digit 1 | MNIST Digit 4 | MNIST Digit 5 | CIFAR-10 | CIFAR-100 |
|---|---|---|---|---|---|
| VAE | $0.072_{\pm0.02}$ | $0.345_{\pm0.03}$ | $0.332_{\pm0.02}$ | $0.158_{\pm0.03}$ | $0.112_{\pm0.02}$ |
| ABP | $0.102_{\pm0.03}$ | $0.142_{\pm0.04}$ | $0.153_{\pm0.03}$ | $0.145_{\pm0.02}$ | $0.107_{\pm0.03}$ |
| MEG | $0.289_{\pm0.04}$ | $0.409_{\pm0.05}$ | $0.415_{\pm0.05}$ | $0.298_{\pm0.04}$ | $0.350_{\pm0.03}$ |
| BiGAN-$\sigma$ | $0.295_{\pm0.03}$ | $0.451_{\pm0.03}$ | $0.520_{\pm0.03}$ | $0.355_{\pm0.03}$ | $0.312_{\pm0.03}$ |
| LEBM | $0.342_{\pm0.02}$ | $0.635_{\pm0.02}$ | $0.622_{\pm0.02}$ | $0.468_{\pm0.02}$ | $0.418_{\pm0.02}$ |
| Adaptive CE | $0.540_{\pm0.02}$ | $0.735_{\pm0.02}$ | $0.748_{\pm0.02}$ | $0.625_{\pm0.02}$ | $0.505_{\pm0.02}$ |
| **M-Star (Ours)** | $\mathbf{0.690_{\pm0.02}}$ | $\mathbf{0.918_{\pm0.01}}$ | $\mathbf{0.942_{\pm0.02}}$ | $\mathbf{0.810_{\pm0.02}}$ | $\mathbf{0.710_{\pm0.02}}$ |

and Inception Score (IS) for diversity and semantic quality. Table 2 summarizes results across resolutions. At $64\times64$, the 8-step student already surpasses the 1024-step teacher in efficiency, and this trend persists at $128\times128$, where it maintains high IS while reducing inference cost. Across resolutions, the alternating optimization variant performs best at low step counts ($k = 8$) and is more stable under extreme budgets ($k = 1$–$2$) than the instant 2-batch variant. Overall, M-Star reduces inference cost while improving robustness by smoothing inconsistent teacher predictions.

### 5.3. How Does Distillation Affect Guidance Sensitivity?

We evaluate the impact of M-Star distillation on student–teacher performance and sensitivity to Classifier-Free Guidance (CFG). As shown in Table 2, the 8-step student consistently outperforms its 512-step teacher, which we attribute to reduced temporal variance: whereas the teacher's long ancestral sampling trajectories accumulate inconsistent intermediate predictions, the distilled student aggregates them into a more stable representation via a tail-based statistic. This can be interpreted as suppressing discretization and drift errors through a denoising jump regularized by a Markovian projection.

**Consistency under Guidance.** M-Star also internalizes CFG during progressive step halving, effectively baking guided behavior into the student through two-step teacher lookaheads. As shown in Figure 4, higher guidance scales improve semantic alignment (IS and CLIP scores) while preserving structural coherence. Unlike standard DDPMs, the hub-and-spoke anchoring prevents the off-manifold drift typically induced by strong guidance.

**Real Geodesic Data.** We evaluate M-Star on a geodesic fire dataset from the Earth's surface (NASA EOSDIS, 2020) by modeling it with a three-dimensional von Mises-Fisher distribution. Following the methodology of SS-DDPM (Okhotin et al., 2023), we generate samples whose distribution closely matches the original data. The resulting samples, alongside the source data, are visualized in Figure 6.

### 5.4. Geodesic and Meteorological Benchmarks

To further demonstrate real-world applicability beyond standard image benchmarks, we evaluate M-Star on three additional datasets that fall outside the Gaussian regime: the

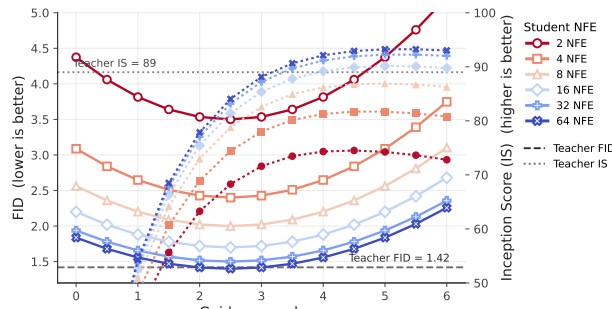

*Figure 4.* **Effect of guidance scale on the distilled M-Star student.** FID (solid, left axis, lower is better) and Inception Score (dotted, right axis, higher is better) as a function of guidance scale $w$, for student models at six NFE budgets (2–64). Dashed and dotted horizontal lines mark the 512-step teacher's FID and IS. All distilled students preserve the U-shaped FID and saturating IS profile of the teacher; the 32-step and 64-step students match or surpass the teacher on both metrics across a wide range of $w$.

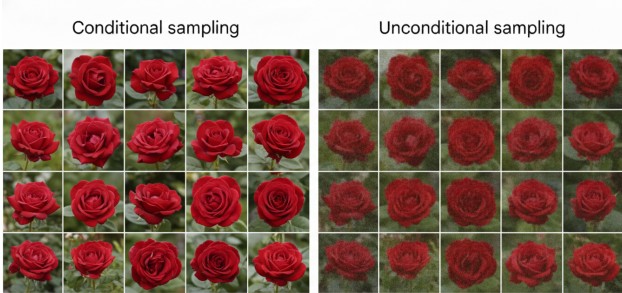

*Figure 5.* ImageNet samples at $128 \times 128$ resolution, generated using the 8-step distilled model with alternating optimization.

NASA EOSDIS wildfire dataset (NASA EOSDIS, 2020) (vMF, $d = 3$); CERRA (Ridal et al., 2024), a pan-European reanalysis at $5.5\,\mathrm{km}$ resolution; and HRES-IFS (ECMWF, 2016), a global forecast system at $9\,\mathrm{km}$ resolution. Both CERRA and HRES-IFS are evaluated on $2\,\mathrm{m}$ temperature. Setup details and metric definitions are provided in Appendix E.6. Table 3 summarizes the results: M-Star outperforms SS-DDPM across all datasets and step budgets. Notably, at 8 steps M-Star degrades only $\sim 7\%$ in CRPS relative to its 1000-step baseline, whereas SS-DDPM degrades by $\sim 40\%$, confirming that the Markovian projection provides step-reduction robustness that SS-DDPM cannot

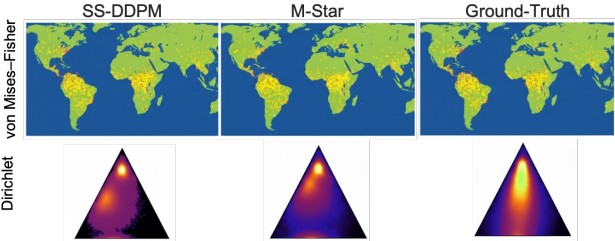

*Figure 6.* The first row shows real data and the second generated samples. For the von Mises–Fisher and Dirichlet models, we show 2D histograms of samples. Pixel intensity reflects ellipse overlap, with darker regions indicating higher density.

*Table 2.* ImageNet results. **(a)** MSE/FID/IS for unconditional generation under vanilla (V) and DAMC (D) sampling; NFE for LEBM-family methods refers to Langevin steps. **(b)** Class-conditional generation on ImageNet $64 \times 64$ and $128 \times 128$; *Base Teacher* is the fully converged 1024-step M-Star prior to distillation, *M-Star – Distillation* is the student obtained by progressive distillation. **(c)** Component ablation at 8 NFE on ImageNet-64. Param counts and Langevin steps for LEBM baselines follow the values reported in their respective.

| Method | # Params | NFE | Sampling | MSE | FID | IS |
|---|---|---|---|---|---|---|
| *(a) Unconditional generation – vanilla vs DAMC sampling* | | | | | | |
| VI-LEBM | 17M | 25 | V | $0.112_{\pm 0.030}$ | $134.5_{\pm 3.2}$ | $18.2_{\pm 0.9}$ |
| VI-LEBM | 17M | 25 | D | $0.094_{\pm 0.025}$ | $128.7_{\pm 3.0}$ | $19.4_{\pm 0.8}$ |
| SR-LEBM | 17M | 25 | V | $0.049_{\pm 0.010}$ | $121.7_{\pm 2.5}$ | $22.1_{\pm 0.7}$ |
| SR-LEBM | 17M | 25 | D | $0.044_{\pm 0.010}$ | $115.3_{\pm 2.3}$ | $23.6_{\pm 0.7}$ |
| DAMC-G | 80M | 100 | V | $0.046_{\pm 0.010}$ | $145.2_{\pm 3.5}$ | $20.5_{\pm 0.9}$ |
| DAMC-G | 80M | 100 | D | $0.039_{\pm 0.010}$ | $119.8_{\pm 2.4}$ | $25.1_{\pm 0.7}$ |
| NALR-LEBM | 75M | 100 | V | $0.058_{\pm 0.020}$ | $118.6_{\pm 2.2}$ | $23.8_{\pm 0.7}$ |
| NALR-LEBM | 75M | 100 | D | $0.040_{\pm 0.010}$ | $112.9_{\pm 2.0}$ | $26.4_{\pm 0.6}$ |
| **M-STAR** | 340M | 8 | V | $0.047_{\pm 0.010}$ | $107.3_{\pm 1.9}$ | $28.7_{\pm 0.6}$ |
| **M-STAR** | 340M | 8 | D | $\mathbf{0.035_{\pm 0.010}}$ | $\mathbf{102.1_{\pm 1.8}}$ | $\mathbf{30.5_{\pm 0.5}}$ |
| *(b) Class-conditional ImageNet $64 \times 64$ – many-step baselines* | | | | | | |
| VDM++ (KINGMA ET AL., 2021) | 2B | 1024 | – | $0.029_{\pm 0.005}$ | $1.43_{\pm 0.04}$ | $64_{\pm 1.2}$ |
| RIN (JABRI ET AL., 2022) | 281M | 1000 | – | $0.026_{\pm 0.004}$ | $\mathbf{1.23_{\pm 0.03}}$ | $67_{\pm 1.1}$ |
| BASE TEACHER (OURS) | 340M | 1024 | – | $0.028_{\pm 0.004}$ | $1.42_{\pm 0.03}$ | $\mathbf{84_{\pm 1.0}}$ |
| *(b) Class-conditional ImageNet $64 \times 64$ – few-step distillation* | | | | | | |
| PD (SALIMANS & HO, 2022) | 400M | 8 | – | $0.041_{\pm 0.007}$ | $1.70_{\pm 0.05}$ | $63_{\pm 1.4}$ |
| MULTISTEP-CD (HEEK ET AL., 2024) | 1.2B | 8 | – | $0.034_{\pm 0.005}$ | $1.40_{\pm 0.04}$ | $73_{\pm 1.2}$ |
| **M-STAR** | 340M | 4 | – | $0.038_{\pm 0.006}$ | $1.50_{\pm 0.04}$ | $75_{\pm 1.1}$ |
| **M-STAR – DISTILLATION** | 340M | 8 | – | $\mathbf{0.030_{\pm 0.005}}$ | $1.24_{\pm 0.03}$ | $78_{\pm 1.0}$ |
| *(b) Class-conditional ImageNet $128 \times 128$* | | | | | | |
| VDM++ (KINGMA ET AL., 2021) | 2B | 1024 | – | $0.034_{\pm 0.006}$ | $1.75_{\pm 0.05}$ | $171_{\pm 2.1}$ |
| BASE TEACHER (OURS) | 340M | 1024 | – | $0.032_{\pm 0.005}$ | $1.76_{\pm 0.05}$ | $\mathbf{194_{\pm 1.8}}$ |
| PD (SALIMANS & HO, 2022) | 400M | 8 | – | $0.049_{\pm 0.008}$ | $2.50_{\pm 0.07}$ | $162_{\pm 2.4}$ |
| MULTISTEP-CD (HEEK ET AL., 2024) | 1.2B | 8 | – | $0.041_{\pm 0.007}$ | $2.10_{\pm 0.06}$ | $160_{\pm 2.3}$ |
| **M-STAR** | 340M | 8 | – | $\mathbf{0.031_{\pm 0.005}}$ | $\mathbf{1.49_{\pm 0.04}}$ | $184_{\pm 1.7}$ |
| *(c) Component ablation – ImageNet-64, 8 NFE* | | | | | | |
| TEACHER (BASE, PROJECTION ONLY) | 340M | 512 | – | $0.028_{\pm 0.004}$ | $1.42_{\pm 0.03}$ | $84_{\pm 1.0}$ |
| SS-DDPM BASELINE (NO PROJECTION, NO DISTILLATION) | 340M | 8 | – | $0.089_{\pm 0.014}$ | $12.45_{\pm 0.42}$ | $41_{\pm 1.8}$ |
| M-STAR (PROJECTION, NO DISTILLATION) | 340M | 8 | – | $0.054_{\pm 0.009}$ | $4.12_{\pm 0.12}$ | $69_{\pm 1.3}$ |
| **M-STAR (PROJECTION + DISTILLATION)** | 340M | 8 | – | $\mathbf{0.053_{\pm 0.009}}$ | $\mathbf{4.10_{\pm 0.11}}$ | $\mathbf{71_{\pm 1.2}}$ |

*Table 3.* Geodesic and meteorological benchmarks at 8 sampling steps. Lower is better for all metrics. GD: great-circle distance; NLL: negative log-likelihood; CRPS: continuous ranked probability score; MMD: RBF kernel. The 1000-step references and a vMF-KDE baseline are reported in Table 13 (App. E.6).

| Dataset | Model | NLL ↓ | GD ↓ (°) | CRPS ↓ | MMD ↓ ($\times 10^{-3}$) |
|---|---|---|---|---|---|
| Wildfire (vMF) | SS-DDPM | $2.58 \pm 0.09$ | $3.02 \pm 0.22$ | – | $4.14 \pm 0.38$ |
| Wildfire (vMF) | **M-Star** | $2.29 \pm 0.06$ | $2.31 \pm 0.17$ | – | $2.89 \pm 0.27$ |
| CERRA (2 m temp.) | SS-DDPM | – | – | $0.589 \pm 0.024$ | $7.83 \pm 0.44$ |
| CERRA (2 m temp.) | **M-Star** | – | – | $0.401 \pm 0.018$ | $4.98 \pm 0.31$ |
| HRES-IFS (2 m temp.) | SS-DDPM | – | – | $0.621 \pm 0.027$ | $8.34 \pm 0.51$ |
| HRES-IFS (2 m temp.) | **M-Star** | – | – | $0.418 \pm 0.019$ | $5.19 \pm 0.33$ |

match and highlights the temporal incoherence issue.

## 6. Conclusion

In this work, we introduced Markovian-Projected Star-Shaped Diffusion (M-Star), a framework that generalizes standard DDPMs to a broader class of exponential family models. Our approach learns a Markovian projection that preserves the marginals of a non-Markovian, star-shaped forward process, thereby addressing key limitations of prior formulations, particularly the loss of temporal consistency in reverse dynamics. The resulting global structure provides stronger coherence across timesteps and makes the model especially well-suited for progressive distillation, enabling efficient few-step and even single-step generation while maintaining high sample quality. Beyond empirical gains, a key contribution of this work is the idea that Markovian projection can act as a principled design tool for constructing structured generative processes. By decoupling the choice of forward dynamics from the constraints of efficient sampling, M-Star opens the door to designing task-specific forward processes that are not restricted by standard diffusion assumptions.

**Limitations.** M-Star relies on linear parameterizations, which may limit performance on complex manifolds, and requires careful tuning of noise schedules and tail statistics. Training and distillation can be computationally intensive for high-resolution data, and extending the method beyond image generation may need further adaptation. The Markovian projection guarantees exact marginal matching by construction, and the ELBO provides a principled training objective. Analyzing the interactions at play during the joint optimization of $(\theta, \mathbf{M}_t)$ appears to be a consequential theoretical challenge left for future works.

## Impact Statement

This work contributes to the field of generative modeling by introducing a versatile diffusion framework. We hope our work unlocks new possibilities in fields that require precise modeling of constrained data (e.g., climate science). While we recognize the potential for generative models to be misused (e.g., misinformation), we believe that the theoretical insights provided here are fundamentally neutral. The eventual impact of this work will be shaped by the values and intentions of those who deploy it in real-world settings.

## Acknowledgments

This work was supported in part by Google Research credits, and by the Hi! PARIS Center through its doctoral grant program. We gratefully acknowledge this support, which contributed to enabling and facilitating the completion of this research. We also thank François Roueff and Randal Douc for their guidance and support throughout this work.

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

# Supplementary Material

## A. Proofs

### A.1. Proof of Theorem 3.4

The result is due to (Okhotin et al., 2023) (Lemma 1 therein); the proof is restated here for completeness and is not a contribution of this paper.

*Proof of Theorem 3.4.* We only need to show that the posterior distribution of the clean data $q(\mathbf{z}_0 \,|\, \mathbf{z}_{t:T})$ depends on the tail $\mathbf{z}_{t:T}$ only through $\mathbf{s}_t$. If this holds, the claim will follow immediately, since

$$q(\mathbf{z}_{t-1} \,|\, \mathbf{z}_{t:T}) = \int q(\mathbf{z}_{t-1} \,|\, \mathbf{z}_0) \, q(\mathbf{z}_0 \,|\, \mathbf{z}_{t:T}) \, d\mathbf{z}_0.$$

Indeed, if $q(\mathbf{z}_0 \,|\, \mathbf{z}_{t:T}) = q(\mathbf{z}_0 \,|\, \mathbf{s}_t)$, then the integral becomes a function of $\mathbf{z}_{t-1}$ and $\mathbf{s}_t$ only.

According to Bayes' rule, the posterior $q(\mathbf{z}_0 \,|\, \mathbf{z}_{t:T})$ writes

$$q(\mathbf{z}_0 \,|\, \mathbf{z}_{t:T}) = \frac{q(\mathbf{z}_{t:T} \,|\, \mathbf{z}_0) \, q(\mathbf{z}_0)}{q(\mathbf{z}_{t:T})}.$$

The samples $(\mathbf{z}_k)_{1 \le s \le T}$ being conditionally independent given $\mathbf{z}_0$ in the non-Markovian framework of (3), the term $q(\mathbf{z}_{t:T} \,|\, \mathbf{z}_0)$ factorizes into $q(\mathbf{z}_{t:T} \,|\, \mathbf{z}_0) = \prod_{k=t}^{T} q(\mathbf{z}_k \,|\, \mathbf{z}_0)$. Substituting the expressions (4) from Assumption 3.2 and (5) from Assumption 3.3, we get

$$q(\mathbf{z}_{t:T} \,|\, \mathbf{z}_0) = \prod_{k=t}^{T} h_k(\mathbf{z}_k) \exp\Big[ \langle \mathbf{A}_s f(\mathbf{z}_0) + \mathbf{b}_s, \psi(\mathbf{z}_k) \rangle - \mathsf{a}_k(\mathbf{z}_0) \Big]$$

$$= \Big( \prod_{k=t}^{T} h_k(\mathbf{z}_k) \Big) \exp\Big[ \sum_{k=t}^{T} \big( f(\mathbf{z}_0)^\top \mathbf{A}_k^\top + \mathbf{b}_k^\top \big) \, \psi(\mathbf{z}_k) - \mathsf{a}_k(\mathbf{z}_0) \Big]$$

$$= C(\mathbf{z}_{t:T}) \exp\Big[ f(\mathbf{z}_0)^\top \Big( \sum_{k=t}^{T} \mathbf{A}_k^\top \, \psi(\mathbf{z}_k) \Big) - \sum_{k=t}^{T} \mathsf{a}_k(\mathbf{z}_0) \Big]$$

$$= C(\mathbf{z}_{t:T}) \exp\Big[ f(\mathbf{z}_0)^\top \mathbf{s}_t - \sum_{k=t}^{T} \mathsf{a}_k(\mathbf{z}_0) \Big],$$

where $C(\mathbf{z}_{t:T})$ groups all the factors that do not depend on $\mathbf{z}_0$. Plugging this back into the posterior expression for $q(\mathbf{z}_0 \,|\, \mathbf{z}_{t:T})$, we get

$$q(\mathbf{z}_0 \,|\, \mathbf{z}_{t:T}) \propto q(\mathbf{z}_0) \exp\Big[ f(\mathbf{z}_0)^\top \mathbf{s}_t - \sum_{k=t}^{T} \mathsf{a}_k(\mathbf{z}_0) \Big].$$

Since the right-hand side is a function of only $\mathbf{z}_0$ and $\mathbf{s}_t$, we have $q(\mathbf{z}_0 \,|\, \mathbf{z}_{t:T}) = q(\mathbf{z}_0 \,|\, \mathbf{s}_t)$. It follows that $q(\mathbf{z}_{t-1} \,|\, \mathbf{z}_{t:T}) = q(\mathbf{z}_{t-1} \,|\, \mathbf{s}_t)$, which completes the proof. $\qquad \square$

### A.2. Proof of Theorem 4.1

We first introduce some notation, and state the assumptions required for the theorem. Let $N = T - t + 1$ denote the length of the tail trajectory. We define $Y_k(\mathbf{z}_0) = \mathbf{A}_k^\top \psi(\mathbf{z}_k)$, so that the sufficient tail statistic writes $\mathbf{s}_t = \sum_{k=t}^{T} Y_k(\mathbf{z}_0)$. Conditioned on $\mathbf{z}_0$ and when the $(\mathbf{z}_k)_{1 \le k \le T}$ are generated by the star-shaped forward process of Assumption 3.1, the terms $(Y_k(\mathbf{z}_0))_{t \le k \le T}$ in this sum are independent random vectors in $\mathbb{R}^d$. Let $\mu_k(\mathbf{z}_0) = \mathbb{E}[Y_k(\mathbf{z}_0)]$ and $\Sigma_k(\mathbf{z}_0) = \mathrm{Cov}(Y_k(\mathbf{z}_0))$.

By independence, the total conditional mean and covariance of the tail statistic are respectively given by

$$\mu_N(\mathbf{z}_0) = \sum_{k=t}^{T} \mu_k(\mathbf{z}_0), \quad B_N(\mathbf{z}_0) = \sum_{k=t}^{T} \Sigma_k(\mathbf{z}_0).$$

We state two assumptions needed in the proof of Theorem 4.1.

**Assumption A.1** (Variance Condition). For all $\mathbf{z}_0 \in \mathbb{R}^d$ such that $q(\mathbf{z}_0) > 0$ and sufficiently large $N$, the total covariance matrix $B_N(\mathbf{z}_0)$ is positive definite.

**Assumption A.2** (Multivariate Lyapunov Condition). For all $\mathbf{z}_0 \in \mathbb{R}^d$ such that $q(\mathbf{z}_0) > 0$, there exists some $\delta > 0$ such that:

$$\lim_{N \to \infty} \sum_{k=t}^{T} \mathbb{E}\left[\left\|B_N(\mathbf{z}_0)^{-1/2}(Y_k(\mathbf{z}_0) - \mu_k(\mathbf{z}_0))\right\|^{2+\delta}\right] = 0.$$

Recall the Lyapunov Central Limit theorem (Billingsley, 2013).

**Theorem A.3.** *For each $N \in \mathbb{N}^*$, let $(X_{N1}, \ldots, X_{NN})$ be independent random variables with zero means and finite variances. Let $S_N = \sum_{k=1}^{N} X_{Nk}$. Suppose that for each $N$, $\sum_{k=1}^{N} \mathrm{Var}(X_{Nk}) = 1$. If there exists $\delta > 0$ such that the Lyapunov condition holds:*

$$\lim_{N \to \infty} \sum_{k=1}^{N} \mathbb{E}[|X_{Nk}|^{2+\delta}] = 0,$$

*then as $N$ goes to infinity we have*

$$S_N \xrightarrow{(d)} \mathcal{N}(0, 1),$$

*where $\xrightarrow{(d)}$ denotes the convergence in distribution.*

We now have all the tools to prove Theorem 4.1.

*Proof of Theorem 4.1.* Since Assumption A.1 guarantees that $B_N(\mathbf{z}_0)$ is a positive definite matrix for sufficiently large $N$, the inverse square root $B_N(z_0)^{-1/2}$ is well-defined. This allows us to write

$$W_N = B_N(\mathbf{z}_0)^{-1/2}(\mathbf{s}_t - \mu_N(\mathbf{z}_0)) = \sum_{k=t}^{T} B_N(\mathbf{z}_0)^{-1/2}(Y_k(\mathbf{z}_0) - \mu_k(\mathbf{z}_0)).$$

To establish asymptotic normality in $\mathbb{R}^d$, we rely on the Cramér-Wold theorem (Cramér & Wold, 1936): the random vector $W_N$ converges in distribution to $\mathcal{N}(0, \mathbf{I})$ in $\mathbb{R}^d$ if and only if for every $u \in \mathbb{S}^{d-1}$, the projection $u^\top W_N$ converges in distribution to $\mathcal{N}(0, 1)$ in $\mathbb{R}$. Here, $\mathbb{S}^{d-1}$ denotes the unit sphere in $\mathbb{R}^d$, i.e., $\mathbb{S}^{d-1} = \{u \in \mathbb{R}^d \ : \ \|u\| = 1\}$.

Consider any $u \in \mathbb{S}^{d-1}$ and any $N \in \mathbb{N}^*$, and let $X_{Nk} = u^\top B_N(\mathbf{z}_0)^{-1/2}(Y_k(\mathbf{z}_0) - \mu_k(\mathbf{z}_0))$, so that $u^\top W_N = \sum_{k=t}^{T} X_{Nk}$ for all $k \in [\![t, T]\!]$. By design, the terms $X_{Nk}$ are independent real random variables with zero mean. Furthermore, the properties of variance imply

$$\mathrm{Var}(u^\top W_N) = \left(u^\top B_N(\mathbf{z}_0)^{-1/2}\right)\left(\sum_{k=t}^{T} \Sigma_k\right)\left(B_N(\mathbf{z}_0)^{-1/2}u\right) = u^\top B_N(\mathbf{z}_0)^{-1/2} B_N(\mathbf{z}_0) B_N(\mathbf{z}_0)^{-1/2}u = u^\top u = 1.$$

To apply Theorem A.3, we need to check that the Lyapunov condition is satisfied. This requires showing that for some $\delta > 0$, we have

$$\lim_{N \to \infty} \sum_{k=t}^{T} \mathbb{E}[|X_{Nk}|^{2+\delta}] = 0.$$

By the Cauchy-Schwarz inequality,

$$|X_{Nk}| = \left|u^\top B_N(\mathbf{z}_0)^{-1/2}(Y_k - \mu_k)\right| \leq \|u\| \left\|B_N(\mathbf{z}_0)^{-1/2}(Y_k - \mu_k)\right\| = \left\|B_N(\mathbf{z}_0)^{-1/2}(Y_k - \mu_k)\right\|.$$

According to Assumption A.2, the right-hand side vanishes as $N$ goes to infinity. Therefore, we can apply Theorem A.3 and conclude that $u^\top W_N$ converges in distribution to a standard Gaussian in $\mathbb{R}$. Finally, by the Cramér-Wold theorem, this implies the multivariate convergence $W_N \xrightarrow{d} \mathcal{N}(0, \mathbf{I})$ in $\mathbb{R}^d$. $\qquad \square$

### A.3. Verification of Assumption A.2 for regular and minimal exponential families

We give a sufficient condition under which the multivariate Lyapunov condition (Assumption A.2) holds. The argument formalizes the heuristic that, for any regular minimal exponential family under a well-posed noise schedule, the high-order moments of $Y_k(\mathbf{z}_0)$ are controlled by $\Sigma_k(\mathbf{z}_0)$.

**Proposition A.4.** *With the notation of Appendix A.2, suppose there exists $\delta > 0$ and a constant $c > 0$ such that*

$$\mathbb{E}\left[\|Y_k(\mathbf{z}_0) - \mu_k(\mathbf{z}_0)\|^{2+\delta}\right] \leq c \operatorname{Tr}(\Sigma_k(\mathbf{z}_0))^{(2+\delta)/2}. \tag{15}$$

*Further assume that there exists a constant $M < \infty$ satisfying $\operatorname{Tr}(\Sigma_k(\mathbf{z}_0)) \leq M$ for all $k \in \{1, \ldots, T\}$, and that the minimum eigenvalue of $B_N$ satisfies $\lambda_{\min}(B_N(\mathbf{z}_0))^{-1} = O(N^{-1})$. Then the Lyapunov condition in Assumption A.2 is satisfied with the same $\delta$.*

*Proof.* We study the limit as $N \to \infty$ of the sum

$$L_N = \sum_{k=t}^{T} \mathbb{E}\left[\|B_N(\mathbf{z}_0)^{-1/2}(Y_k(\mathbf{z}_0) - \mu_k(\mathbf{z}_0))\|^{2+\delta}\right].$$

By sub-multiplicativity of the operator norm, we have:

$$\|B_N(\mathbf{z}_0)^{-1/2}(Y_k(\mathbf{z}_0) - \mu_k(\mathbf{z}_0))\| \leq \|B_N(\mathbf{z}_0)^{-1/2}\|_{\text{op}}\|Y_k(\mathbf{z}_0) - \mu_k(\mathbf{z}_0)\|.$$

The operator norm of $B_N(\mathbf{z}_0)^{-1/2}$ is exactly $\lambda_{\min}(B_N(\mathbf{z}_0))^{-1/2}$. Substituting (15), we get:

$$L_N \leq \lambda_{\min}(B_N(\mathbf{z}_0))^{-(2+\delta)/2} \sum_{k=t}^{T} \mathbb{E}\left[\|Y_k(\mathbf{z}_0) - \mu_k(\mathbf{z}_0)\|^{2+\delta}\right] \leq c\,\lambda_{\min}(B_N(\mathbf{z}_0))^{-(2+\delta)/2} \sum_{k=t}^{T} \operatorname{Tr}(\Sigma_k(\mathbf{z}_0))^{(2+\delta)/2}.$$

Splitting the exponent in the sum and using the upper-bound in $M$ yields

$$\sum_{k=t}^{T} \operatorname{Tr}(\Sigma_k(\mathbf{z}_0))^{(2+\delta)/2} \leq M^{\delta/2} \sum_{k=t}^{T} \operatorname{Tr}(\Sigma_k(\mathbf{z}_0)) = M^{\delta/2} \operatorname{Tr}(B_N(\mathbf{z}_0)).$$

Substituting this back into our bound for $L_N$, we obtain

$$L_N \leq c\,M^{\delta/2} \frac{\operatorname{Tr}(B_N(\mathbf{z}_0))}{\lambda_{\min}(B_N(\mathbf{z}_0))^{(2+\delta)/2}}.$$

By assumption, $\operatorname{Tr}(B_N(\mathbf{z}_0)) = O(N)$ and $\lambda_{\min}(B_N(\mathbf{z}_0))^{-1} = O(N^{-1})$. Therefore, the asymptotic behavior of the bound is

$$L_N = O\left(\frac{N}{N^{1+\delta/2}}\right) = O(N^{-\delta/2}).$$

Since $\delta > 0$, the Lyapunov condition in Assumption A.2 is satisfied. $\qquad\square$

Exponential families that satisfy Assumption 3.2 are minimal and regular, hence the covariance matrices $\Sigma_k(\mathbf{z}_0)$ are Hessian matrices of the strictly convex and smooth log-partition function. Assuming the natural parameters remains confined to a compact subset in the interior of $E$ (as is the case if we assume a well-posed noise schedule), the eigenvalues of $\Sigma_k(\mathbf{z}_0)$ are uniformly bounded from above and strictly bounded away from zero. Therefore, the assumptions that $\operatorname{Tr}(\Sigma_k(\mathbf{z}_0)) \leq M$ for all $k \in \{1, \ldots, T\}$ and some $M > 0$, and that $\lambda_{\min}(B_N(\mathbf{z}_0))^{-1} = O(N^{-1})$ are both satisfied. As for the moment condition, regular exponential families have moments at all orders, given by the derivatives of the $\mathcal{C}^{\infty}$ log-partition function. These moments are also uniformly bounded over the same compact set, meaning that (15) holds.

# B. On the Interaction Parameter

## B.1. Validity of the kernel form (9)

In (9), we let

$$k_t(\mathbf{z}_t \mid \mathbf{z}_{t-1}, \mathbf{z}_0) = q(\mathbf{z}_t \mid \mathbf{z}_0) \frac{\pi_t(\mathbf{z}_{t-1} \mid \mathbf{z}_t, \mathbf{z}_0)}{q(\mathbf{z}_{t-1} \mid \mathbf{z}_0)},$$

where $\int \pi_t(\mathbf{z}_{t-1} \mid \mathbf{z}_t, \mathbf{z}_0) \, \nu(\mathrm{d}\mathbf{z}_{t-1}) = 1$. Plug this expression into the right-hand side of (7):

$$\int k_t(\mathbf{z}_t \mid \mathbf{z}_{t-1}, \mathbf{z}_0) \, q(\mathbf{z}_{t-1} \mid \mathbf{z}_0) \, \nu(\mathrm{d}\mathbf{z}_{t-1}) = \int q(\mathbf{z}_t \mid \mathbf{z}_0) \frac{\pi_t(\mathbf{z}_{t-1} \mid \mathbf{z}_t, \mathbf{z}_0)}{q(\mathbf{z}_{t-1} \mid \mathbf{z}_0)} q(\mathbf{z}_{t-1} \mid \mathbf{z}_0) \, \nu(\mathrm{d}\mathbf{z}_{t-1})$$

$$= q(\mathbf{z}_t \mid \mathbf{z}_0) \int \pi_t(\mathbf{z}_{t-1} \mid \mathbf{z}_t, \mathbf{z}_0) \, \nu(\mathrm{d}\mathbf{z}_{t-1})$$

$$= q(\mathbf{z}_t \mid \mathbf{z}_0).$$

Therefore, setting $k_t$ as in (9) with any probability density function $\pi_t$ guarantees that the Chapman-Kolmogorov equation (7) is satisfied.

## B.2. Kullback-Leibler divergence between exponential family distributions as a Bregman divergence

The objective in (12) can be optimized easily since, under Assumptions 3.2 and 3.3, the densities $\tilde{q}$ belong to the same regular minimal exponential family. In what follows, we give the simple formula we use to compute the objective function and its gradients.

**Definition B.1** (Bregman Divergence). Let $\Omega \subseteq \mathbb{R}^d$ be an open convex set, and let $F : \Omega \to \mathbb{R}$ be a strictly convex, continuously differentiable function. The Bregman divergence $d_F : \Omega^2 \to \mathbb{R}_+$ induced by $F$ is defined for $(\mathbf{x}, \mathbf{y}) \in \Omega^2$ by

$$d_F(\mathbf{x}, \mathbf{y}) = F(\mathbf{x}) - F(\mathbf{y}) - \langle \nabla F(\mathbf{y}), \mathbf{x} - \mathbf{y} \rangle.$$

**Proposition B.2.** *Let $q_1$ and $q_2$ be two densities from the same regular and minimal exponential family with sufficient statistic $\psi$, log-partition function $\mathsf{a}$, and natural parameters $\eta_1$ and $\eta_2$ in $E = \{\eta \in \mathbb{R}^d \ : \ |\mathsf{a}(\eta)| < +\infty\}$, that is,*

$$q_1(\mathbf{z}) = \exp[\langle \eta_1, \psi(\mathbf{z}) \rangle - \mathsf{a}(\eta_1)] \quad and \quad q_2(\mathbf{z}) = \exp[\langle \eta_2, \psi(\mathbf{z}) \rangle - \mathsf{a}(\eta_2)].$$

*Then, the Kullback-Leibler divergence between $q_1$ and $q_2$ writes as the Bregman divergence induced by $\mathsf{a}$ between $\eta_2$ and $\eta_1$:*

$$D_{\mathrm{KL}}(q_1 \| q_2) = d_{\mathsf{a}}(\eta_2, \eta_1).$$

*Proof.* It is a well-known fact that the set $E$ is convex (Wainwright & Jordan, 2008). Moreover, an exponential family is regular if $E$ is open, and if it is minimal then $\mathsf{a}$ is a strictly convex function, and its gradient $\nabla \mathsf{a}$ is a smooth diffeomorphism. Hence, the Bregman divergence $d_{\mathsf{a}}$ is well-defined. We have:

$$D_{\mathrm{KL}}(q_1 \| q_2) = \int \log\left(\frac{q_1(\mathbf{z})}{q_2(\mathbf{z})}\right) q_1(\mathbf{z}) \, \nu(\mathrm{d}\mathbf{z})$$

$$= \mathbb{E}_{p_1}[\langle \eta_1, \psi(\mathbf{z}) \rangle - \mathsf{a}(\eta_1) - \langle \eta_2, \psi(\mathbf{z}) \rangle + \mathsf{a}(\eta_2)]$$

$$= \langle \eta_1, \nabla \mathsf{a}(\eta_1) \rangle - \mathsf{a}(\eta_1) - \langle \eta_2, \nabla \mathsf{a}(\eta_1) \rangle + \mathsf{a}(\eta_2)$$

$$= \mathsf{a}(\eta_2) - \mathsf{a}(\eta_1) - \langle \nabla \mathsf{a}(\eta_1), \eta_2 - \eta_1 \rangle$$

$$= d_{\mathsf{a}}(\eta_2, \eta_1).$$

In the second line, we use the classical identity $\mathbb{E}_{q_1}[\psi] = \nabla \mathsf{a}(\eta_1)$. The result follows by reorganizing the terms of the equation. $\qquad\square$

## B.3. Posterior Matching between M-Star and the standard Gaussian DDPM

We derive the natural parameter of the Gaussian density with mean and covariance matrix respectively given by (1) and (2). This shows that the density $q_{\mathrm{M}}(\mathbf{z}_{t-1} \mid \mathbf{z}_t, \mathbf{z}_0)$ in the Gaussian DDPM framework of Section 3.1 matches the posterior

$\tilde{q}(\mathbf{z}_{t-1} \mid \mathbf{z}_t, \mathbf{z}_0)$ of the M-Star model under the assumptions explicited in Section 4.4. As per (11), the density $\tilde{q}(\mathbf{z}_{t-1} \mid \mathbf{z}_t, \mathbf{z}_0)$ is Gaussian with natural parameter

$$\tilde{\eta}_{t-1}(\mathbf{z}_0, \mathbf{z}_t) = \eta_{t-1}(\mathbf{z}_0) + \mathbf{M}_t \psi(\mathbf{z}_t) = \frac{\sqrt{\bar{\alpha}_{t-1}}}{1 - \bar{\alpha}_{t-1}} \mathbf{z}_0 + \frac{\sqrt{\alpha_t}}{\beta_t} \mathbf{z}_t.$$

Using the expressions in (1) and (2), and the noise schedule notation of Section 3.1, we have

$$\tilde{\mu}_t(\mathbf{z}_t, \mathbf{z}_0) = \frac{\sqrt{1 - \beta_t}(1 - \bar{\alpha}_{t-1})}{1 - \bar{\alpha}_t} \mathbf{z}_t + \frac{\sqrt{\bar{\alpha}_{t-1}}\beta_t}{1 - \bar{\alpha}_t} \mathbf{z}_0, \quad \text{and} \quad \tilde{\beta}_t = \frac{1 - \bar{\alpha}_{t-1}}{1 - \bar{\alpha}_t} \beta_t.$$

Therefore, the natural parameter of the DDPM posterior is given by

$$
\begin{aligned}
\tilde{\beta}_t \tilde{\mu}_t(\mathbf{z}_t, \mathbf{z}_0) &= \frac{1 - \bar{\alpha}_t}{1 - \bar{\alpha}_{t-1}} \frac{1}{\beta_t} \left[ \frac{\sqrt{1 - \beta_t}(1 - \bar{\alpha}_{t-1})}{1 - \bar{\alpha}_t} \mathbf{z}_t + \frac{\sqrt{\bar{\alpha}_{t-1}}\beta_t}{1 - \bar{\alpha}_t} \mathbf{z}_0 \right] \\
&= \frac{\sqrt{1 - \beta_t}}{\beta_t} \mathbf{z}_t + \frac{\sqrt{\bar{\alpha}_{t-1}}}{1 - \bar{\alpha}_{t-1}} \mathbf{z}_0 \\
&= \underbrace{\frac{\sqrt{\bar{\alpha}_{t-1}}}{1 - \bar{\alpha}_{t-1}} \mathbf{z}_0}_{\eta_{t-1}(\mathbf{z}_0)} + \underbrace{\frac{\sqrt{\alpha_t}}{\beta_t}}_{\mathbf{M}_t} \mathbf{z}_t \\
&= \tilde{\eta}_{t-1}(\mathbf{z}_0, \mathbf{z}_t).
\end{aligned}
$$

We retrieve the same natural parameter, meaning both distributions are equal.

## C. Example of derivations with the von Mises-Fisher distribution

The von Mises-Fisher (vMF) distribution is supported on the unit sphere $\mathbb{S}^{d-1} = \{\mathbf{z} \in \mathbb{R}^d \; : \; \|\mathbf{z}\| = 1\}$. It is particularly useful in the context of directional statistics, or for any type of spherical data, e.g., geodesic data. If $\nu$ denotes the uniform probability measure on $\mathbb{S}^{d-1}$, the vMF distribution admits a density $\text{vMF}(\,\cdot\,; \eta)$ with respect to $\nu$, characterized by $\text{vMF}(\mathbf{z}; \eta) \propto \exp(\langle \eta, \mathbf{z} \rangle)$, where $\eta \in \mathbb{R}^d$. Usually, we let $\eta = \kappa\mu$, where $\mu \in \mathbb{S}^{d-1}$ is the mean direction and $\kappa \in (0, +\infty)$ is the concentration parameter. The sufficient statistic is clearly $\psi(\mathbf{z}) = \mathbf{z}$.

### C.1. General design

**Marginal distributions.** We define the forward process such that the marginals are vMF distributions centered at the clean data $\mathbf{z}_0$ with a concentration schedule $(\kappa_t)$:

$$q(\mathbf{z}_t \mid \mathbf{z}_0) = \text{vMF}(\mathbf{z}_t; \mathbf{z}_0, \kappa_t) \propto \exp\big(\kappa_t \mathbf{z}_0^\top \mathbf{z}_t\big).$$

With $\mathbf{A}_t = \kappa_t \mathbf{I}$ and $\mathbf{b}_t = 0$, this satisfies Assumption 3.3. As $\kappa_t \to +\infty$, the distribution concentrates around $\mathbf{z}_0$, and as $\kappa_t \to 0$, it becomes the uniform distribution on the sphere. Hence, boundary conditions can easily be satisfied.

**Noise schedule.** Following the heuristics offered by Theorem 4.1, we find a simple approximation of the SNR of the sufficient tail statistic $\mathsf{s}_t$ in the high-concentration regime ($\kappa_t \gg 1$, i.e., $t$ close to zero). Given a variance schedule $(\bar{\alpha}_t)$ designed for a Gaussian DDPM, computations described below

$$\kappa_t = (d - 1) \left( \frac{\bar{\alpha}_t}{1 - \bar{\alpha}_t} - \frac{\bar{\alpha}_{t+1}}{1 - \bar{\alpha}_{t+1}} \right).$$

**Sampling distribution.** The natural parameter for the density $\tilde{q}(\mathbf{z}_{t-1} \mid \mathbf{z}_t, \mathbf{z}_0)$ is $\tilde{\eta}_{t-1}(\mathbf{z}_0, \mathbf{z}_t) = \kappa_{t-1}\mathbf{z}_0 + \mathbf{M}_t \mathbf{z}_t$. We optimize the matrix $\mathbf{M}_t$ as per (12).

### C.2. Noise schedule derivation

First, we give an approximation of the SNR of the sufficient statistic in the high-concentration regime (i.e., when $N = T + t - 1$ is large, close to the data). Then, we match this expression with the SNR yielded by a given noise schedule in a Gaussian DDPM.

**Moments of the von Mises-Fisher distribution in the high-concentration regime.** It is known (Shyntar & Hillen, 2025) that if $\mathbf{z} \sim \mathrm{vMF}(\,\cdot\,; \kappa, \mu)$, then we have:

$$\mathbb{E}[\mathbf{z}] = \frac{I_{d/2}(\kappa)}{I_{d/2-1}(\kappa)}\,\mu \quad \text{and} \quad \mathrm{Cov}(\mathbf{z}) = \frac{I_{d/2}(\kappa)}{\kappa I_{d/2-1}(\kappa)}\,\mathbf{I} + \left[\frac{I_{d/2+1}(\kappa)}{I_{d/2-1}(\kappa)} - \left(\frac{I_{d/2}(\kappa)}{I_{d/2-1}(\kappa)}\right)^2\right]\mu\mu^\top,$$

where $I_\alpha$ denotes the modified Bessel function of the first kind of order $\alpha$.

Let us compute the trace of the covariance matrix. Using the linearity of the trace operator, we treat each term individually. In $\mathbb{R}^d$, the trace of the identity matrix is $d$. For the second term, we use the property that the trace is invariant to transposition along with the fact that $\mu$ lies on the hypersphere $\mathbb{S}^{d-1}$. We deduce

$$\mathrm{Tr}(\mathrm{Cov}(\mathbf{z})) = d\,\frac{I_{d/2}(\kappa)}{\kappa I_{d/2-1}(\kappa)} + \left[\frac{I_{d/2+1}(\kappa)}{I_{d/2-1}(\kappa)} - \left(\frac{I_{d/2}(\kappa)}{I_{d/2-1}(\kappa)}\right)^2\right].$$

We then use the standard recurrence relation for modified Bessel functions of the first kind:

$$I_{\alpha-1}(\kappa) - I_{\alpha+1}(\kappa) = \frac{2\alpha}{\kappa}I_\alpha(\kappa).$$

Setting $\alpha = d/2$, we have:

$$I_{d/2-1}(\kappa) - I_{d/2+1}(\kappa) = \frac{n}{\kappa}I_{d/2}(\kappa).$$

Rearranging the terms and dividing by $I_{d/2-1}(\kappa) > 0$ yields

$$I_{d/2+1}(\kappa) = I_{d/2-1}(\kappa) - \frac{d}{\kappa}I_{d/2}(\kappa), \quad \text{and thus} \quad \frac{I_{d/2+1}(\kappa)}{I_{d/2-1}(\kappa)} = 1 - \frac{dI_{d/2}(\kappa)}{\kappa I_{d/2-1}(\kappa)}.$$

Substitute this expression back into our expression for the trace:

$$\mathrm{Tr}(\mathrm{Cov}(\mathbf{z})) = \frac{dI_{d/2}(\kappa)}{\kappa I_{d/2-1}(\kappa)} + \left(1 - \frac{dI_{d/2}(\kappa)}{\kappa I_{d/2-1}(\kappa)}\right) - \left(\frac{I_{d/2}(\kappa)}{I_{d/2-1}(\kappa)}\right)^2 = 1 - \left(\frac{I_{d/2}(\kappa)}{I_{d/2-1}(\kappa)}\right)^2.$$

Under the high-concentration assumption, i.e., when $\kappa \gg 1$), we have the approximation

$$\frac{I_{d/2}(\kappa)}{I_{d/2-1}(\kappa)} \underset{\kappa \gg 1}{\approx} 1 - \frac{d-1}{2\kappa}.$$

Assuming the term in $1/\kappa^2$ is negligible, this yields

$$\mathrm{Tr}(\mathrm{Cov}(\mathbf{z})) \underset{\kappa \gg 1}{\approx} \frac{d-1}{\kappa}.$$

With the same tools, we also get the approximation

$$\mathbb{E}[\mathbf{z}] \underset{\kappa \gg 1}{\approx} \left(1 - \frac{d-1}{2\kappa}\right)\mu \underset{\kappa \gg 1}{\approx} \mu.$$

**Approximate moments of the tail statistic.** We can now approximate the moments of the sufficient tail statistic. Recall that in the context of Appendix C, we have $\mathbf{s}_t = \sum_{k=t}^T \kappa_k \mathbf{z}_k$. Since the samples $\mathbf{z}_k$ are conditionally independent given $\mathbf{z}_0$,

$$\mathbb{E}[\mathbf{s}_t \mid \mathbf{z}_0] = \left(\sum_{k=t}^T \kappa_k \mathbb{E}_{q(\mathbf{z}_k \mid \mathbf{z}_0)}[\mathbf{z}_k]\right) \underset{T \gg t}{\approx} \left(\sum_{k=t}^T \kappa_k\right)\mathbf{z}_0,$$

and

$$\mathrm{Tr}(\mathrm{Cov}(\mathbf{s}_t \mid \mathbf{z}_0)) = \sum_{k=t}^T \kappa_k^2\,\mathrm{Tr}(\mathrm{Cov}_{q(\mathbf{z}_k \mid \mathbf{z}_0)}(\mathbf{z}_k)) \underset{T \gg t}{\approx} \sum_{k=t}^T \kappa_k^2\left(\frac{d-1}{\kappa_k}\right) = (d-1)\sum_{k=t}^T \kappa_k.$$

Note that while the approximation is only valid for large $\kappa_k$, the smaller values do not contribute to the sum as much and thus it still makes sense to write these approximations.

**SNR matching.** Recall that the SNR of the tail statistic is defined as

$$\text{SNR}(\mathsf{s}_t) = \frac{\|\mathbb{E}[\mathsf{s}_t \mid \mathbf{z}_0]\|^2}{\text{Tr}(\text{Cov}(\mathsf{s}_t \mid \mathbf{z}_0))}.$$

Letting $S_t = \sum_{k=t}^{T} \kappa_k$, we obtain the approximation:

$$\text{SNR}(\mathsf{s}_t) \underset{T \gg t}{\approx} \frac{S_t^2}{(d-1)S_t} = \frac{S_t}{d-1}.$$

We match this to the SNR of a Gaussian DDPM with variance schedule $(\bar{\alpha}_t)$, given by $\text{SNR}_{\text{DDPM}}(t) = \bar{\alpha}_t/(1-\bar{\alpha}_t)$:

$$\frac{S_t}{d-1} = \frac{\bar{\alpha}_t}{1-\bar{\alpha}_t}.$$

Since $S_t = \kappa_t + S_{t+1}$, it follows that

$$\kappa_t = (d-1)\left(\frac{\bar{\alpha}_t}{1-\bar{\alpha}_t} - \frac{\bar{\alpha}_{t+1}}{1-\bar{\alpha}_{t+1}}\right).$$

For the final step $t = T$, we simply assume $\bar{\alpha}_{T+1} = 0$. This formula directly provides a concentration schedule $(\kappa_t)$ that allows to approximate the behavior induced by a given noise schedule in a Gaussian DDPM, in terms of SNR. While the approximation is weaker when $t$ is closer to $T$, this corresponds to a very noisy part of the diffusion process which is less important to the final sample quality in practice and is skipped almost entirely in distillation.

## D. Practical Considerations

**Network architecture Implementation.** The network $\mathbf{z}_\theta$ is trained to predict the data $\mathbf{z}_0$ from the statistic $\mathsf{s}_t$ and the timestep $t$. If the data $\mathbf{z}_0$ is constrained to a specific manifold (e.g., the unit sphere for geodesic data or the probability simplex for compositional data), the network's raw output must be projected onto that manifold. This is typically done via the final layer of the network, for instance, using $L^2$ normalization to project onto a sphere or a softmax function to project onto the simplex.

**Loss reweighting.** The terms in the ELBO objective can have very different magnitudes across timesteps. As is common practice in the literature, we find that training can be stabilized by reweighting the terms of the loss function. This prevents the model from focusing too heavily on specific timesteps (e.g., those with very low noise), leading to better overall sample quality.

**Why Progressive Distillation.** Progressive Distillation (PD) is the natural fit because M-Star's objective reconstructs $\mathbf{z}_0$ from $\mathsf{s}_t$ at all timesteps, matching what PD exploits: teacher and student predict the same target from statistics of different tail lengths. To verify that the gain originates from the Markovian projection rather than the distillation algorithm, we replace PD with Shortcut conditioning (Frans et al., 2025) and observe a comparable improvement over SS-DDPM (Table 4). Consistency models (Song et al., 2023) would require a probability-flow ODE which is generally unavailable for discrete exponential families; we leave this extension to future work.

*Table 4.* Distillation algorithm vs. Markovian projection. The $-8.3$ FID gain is preserved under Shortcut conditioning, confirming that the improvement originates from M-Star's structural properties rather than from the choice of distillation procedure.

| METHOD | NFE | FID $\downarrow$ |
|---|---|---|
| SS-DDPM + PD | 8 | $12.45 \pm 0.42$ |
| M-STAR + SHORTCUT (FRANS ET AL., 2025) | 8 | $4.31 \pm 0.22$ |
| **M-STAR + PD (OURS)** | **8** | $\mathbf{4.12 \pm 0.18}$ |

**Implementation and Computational Cost.** Because we learn a separate diagonal matrix for each timestep, M-Star effectively adds new parameters in the form of an array of size $T \times d$. For high-dimensional state spaces and a large number

of timesteps, it is viable to parameterize these matrices with a lightweight neural network and limit the memory overhead. The added computational cost of learning $\mathbf{M}_t$ is low, since the loss in (12) does not require evaluating the U-Net, $\mathbf{z}_\theta$. Table 5 reports the overhead measured on ImageNet-64 jointly with the FID/IS ablation: the fixed analytical $\mathbf{M}_t$ from Equation (14) is essentially free ($+0.4\%$ FLOPs) and already closes most of the gap with SS-DDPM, while the fully learned $\mathbf{M}_t$ adds only $\sim 8$ minutes over a 68-hour run ($+4.2\%$ FLOPs, $<3\%$ memory) and yields a $-8.3$ FID gain at 8 steps. The gap between fixed and learned widens at low step counts, confirming that the learned interaction matrix better adapts the trajectory for distillation.

*Table 5.* $\mathbf{M}_t$ ablation on ImageNet-64: jointly reporting computational overhead and sample quality. Setting $\mathbf{M}_t = 0$ recovers SS-DDPM; the fixed analytical form (14) closes most of the gap at $+0.4\%$ FLOPs; the learned variant refines it further at $+4.2\%$ FLOPs ($\sim 8$ min over a 68 h run, $< 3\%$ memory).

| $\mathbf{M}_t$ STRATEGY | EXTRA FLOPS | GPU MEM (GB) | FID $\downarrow$ (8) | IS $\uparrow$ (8) | FID $\downarrow$ (1000) | IS $\uparrow$ (1000) |
|---|---|---|---|---|---|---|
| $\mathbf{M}_t = 0$ (SS-DDPM) | – | $38.1 \pm 0.2$ | $12.45 \pm 0.42$ | $51.3 \pm 1.4$ | $3.87 \pm 0.18$ | $71.2 \pm 1.1$ |
| FIXED ANALYTICAL (EQ. 14) | $+0.4\%$ | $38.3 \pm 0.2$ | $4.38 \pm 0.21$ | $73.1 \pm 0.9$ | $1.44 \pm 0.07$ | $83.6 \pm 0.8$ |
| LEARNED $\mathbf{M}_t$ (M-STAR) | $+4.2\%$ | $38.9 \pm 0.2$ | $\mathbf{4.12 \pm 0.18}$ | $\mathbf{74.5 \pm 0.8}$ | $\mathbf{1.42 \pm 0.06}$ | $\mathbf{84.1 \pm 0.7}$ |

# E. Experiments

## E.1. Implementation Details and Hyperparameters

To ensure reproducibility and enable a rigorous comparison with established distillation baselines, we detail the optimization settings used in the M-Star pipeline. We follow the architectural configuration of the ADM backbone (Dhariwal & Nichol, 2021), while adapting the training procedure to our Markovian projection framework.

**Architecture and Noise Schedule.** For ImageNet-64 and ImageNet-128, we employ a U-Net architecture with channel multipliers $\{1, 2, 3, 4\}$ and a constant dropout rate of $0.1$. The teacher model is trained using a linear variance schedule $\beta_t$ over 1000 diffusion steps. Consistent with our star-shaped formulation, the noise levels $\sigma_t$ are defined such that the signal-to-noise ratio (SNR) decays monotonically toward the noise tail.

**Progressive Distillation Settings.** At each halving stage of the distillation process, the student model is initialized with the parameters of the current teacher. We find that using a lower learning rate than that of the initial pretraining phase helps prevent catastrophic forgetting of the hub-reconstruction capability. In addition to the mean-squared error (MSE) objective, we incorporate a perceptual LPIPS loss (Zhang et al., 2018) with weight $\lambda_{\text{LPIPS}}$ to preserve high-frequency details in low-step regimes.

*Table 6.* Hyperparameter configurations for M-Star progressive distillation.

| HYPERPARAMETER | |
|---|---|
| OPTIMIZER | ADAM |
| LEARNING RATE | $2 \times 10^{-4}$ |
| ADAM ($\beta_1, \beta_2$) | $(0.9, 0.999)$ |
| BATCH SIZE | 512 |
| EMA DECAY | 0.9999 |
| LPIPS WEIGHT ($\lambda_{\text{LPIPS}}$) | 0.05 |
| STEPS PER STAGE | 50,000 |
| MIXED PRECISION | FP16 |
| GUIDANCE SCALE ($s$) | 1.0–7.5 |

**Optimization and Hardware.** We use the Adam optimizer with a batch size of 512, distributed across $8\times$ NVIDIA H100 GPUs. For training stability, we apply an Exponential Moving Average (EMA) to the student parameters with a decay rate of 0.9999. The total distillation cost for ImageNet-64 across all stages is approximately 72 GPU-hours.

## E.2. Ablation study on CIFAR datasets

We present an ablation study on CIFAR-10. We compare M-Star (M-Star) with various baseline setups using vanilla (V) and DAMC (D) sampling. M-Star consistently achieves the lowest FID and competitive MSE, demonstrating the benefits of

*Table 7.* Ablation study on CIFAR-100 dataset. VI denotes learning LEBM using variational methods. Results are provided using vanilla (V) and DAMC (D) sampling; $\pm$ denotes standard deviation.

| Model | MSE | FID |
|---|---|---|
| VI-LEBM (V) | $0.084 \pm 0.02$ | $102.3 \pm 2.1$ |
| VI-LEBM (D) | – | – |
| SR-LEBM (V) | $0.038 \pm 0.01$ | $93.7 \pm 1.9$ |
| SR-LEBM (D) | – | – |
| DAMC-G (V) | $0.035 \pm 0.01$ | $108.4 \pm 2.3$ |
| DAMC-G (D) | $0.029 \pm 0.01$ | $88.1 \pm 1.7$ |
| NALR-LEBM (V) | $0.045 \pm 0.02$ | $90.2 \pm 1.8$ |
| NALR-LEBM (D) | $0.028 \pm 0.01$ | $85.6 \pm 1.5$ |
| **M-Star (V)** | $0.036 \pm 0.01$ | $82.7 \pm 1.4$ |
| **M-Star (D)** | $0.025 \pm 0.01$ | $79.4 \pm 1.3$ |

*Table 8.* Ablation study on CIFAR-10 dataset. VI denotes learning LEBM using variational methods. Results are provided using vanilla (V) and DAMC (D) sampling; $\pm$ denotes standard deviation.

| Model | MSE | FID |
|---|---|---|
| VI-LEBM (V) | $0.056 \pm 0.01$ | $79.2 \pm 1.5$ |
| VI-LEBM (D) | – | – |
| SR-LEBM (V) | $0.022 \pm 0.01$ | $71.8 \pm 1.3$ |
| SR-LEBM (D) | – | – |
| DAMC-G (V) | $0.020 \pm 0.01$ | $91.4 \pm 1.7$ |
| DAMC-G (D) | $0.017 \pm 0.01$ | $67.9 \pm 1.2$ |
| NALR-LEBM (V) | $0.030 \pm 0.01$ | $69.5 \pm 1.4$ |
| NALR-LEBM (D) | $0.018 \pm 0.01$ | $65.2 \pm 1.1$ |
| **M-Star (V)** | $0.023 \pm 0.01$ | $61.5 \pm 1.0$ |
| **M-Star (D)** | $0.016 \pm 0.01$ | $58.3 \pm 1.0$ |

heavy-tailed diffusion modeling prior to application to weather data.

### E.3. Convergence of the Interaction Matrix in the Gaussian Case

We provide additional examples of the convergence of the M-Star reverse posterior to the DDPM-implied posterior under Gaussian transitions.

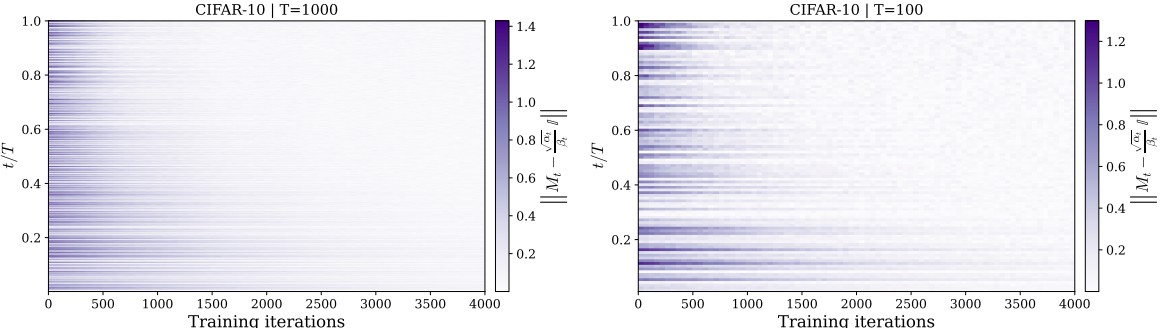

*Figure 7.* Heatmap of the convergence error $\|\mathbf{M}_t - \frac{\sqrt{\alpha_t}}{\beta_t}\mathbf{I}\|$ over training iterations and diffusion timesteps $t/T$ on CIFAR-10 for $T = 100$ and $T = 1000$. Late timesteps show slower convergence and higher residual error.

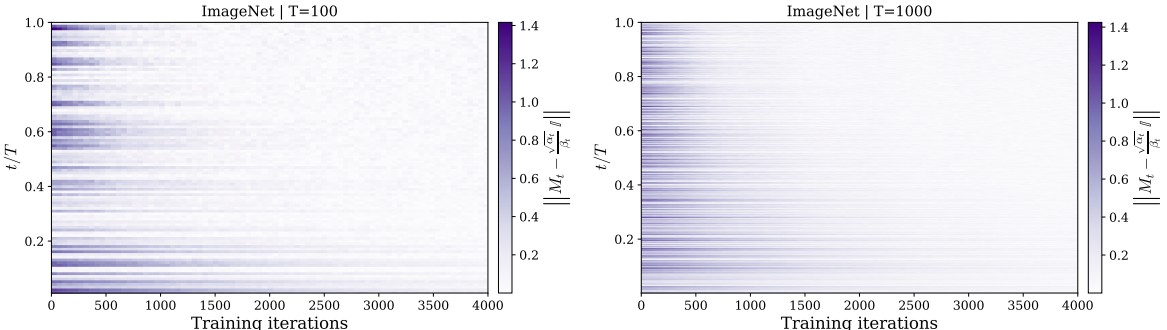

*Figure 8.* Heatmap of the convergence error $\|\mathbf{M}_t - \frac{\sqrt{\alpha_t}}{\beta_t}\mathbf{I}\|$ over training iterations and diffusion timesteps $t/T$ on ImageNet for $T = 100$ and $T = 1000$. Late timesteps show slower convergence and higher residual error.

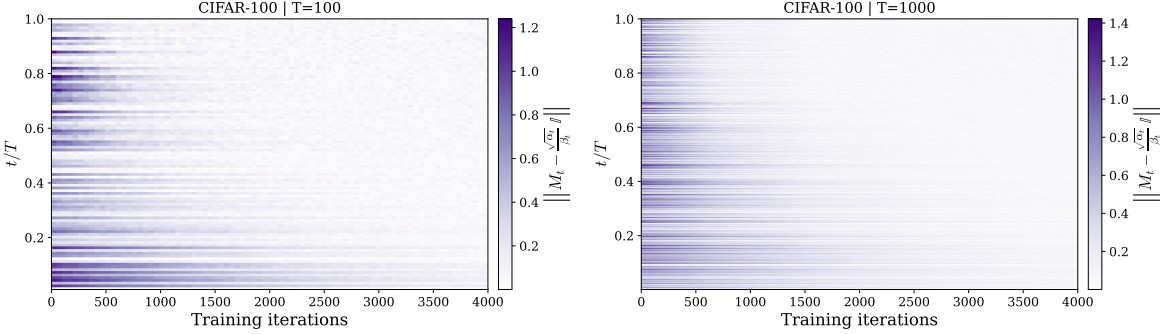

*Figure 9.* Heatmap of the convergence error $\|\mathbf{M}_t - \frac{\sqrt{\alpha_t}}{\beta_t}\mathbf{I}\|$ over training iterations and diffusion timesteps $t/T$ on CIFAR-100 for $T = 100$ and $T = 1000$. Late timesteps show slower convergence and higher residual error.

Figures 7–9 show the convergence error $\|\mathbf{M}_t - \frac{\sqrt{\alpha_t}}{\beta_t}\mathbf{I}\|$ across training iterations and diffusion timesteps $t/T$ for CIFAR-10, ImageNet, and CIFAR-100 at $T = 100$ and $T = 1000$. Early timesteps converge quickly with low residuals, while late timesteps converge slower and retain higher errors, reflecting the increased difficulty of modeling long-range dependencies. Increasing the total timesteps $T$ slows convergence overall, especially at later steps. ImageNet exhibits higher residuals than CIFAR datasets due to greater complexity, yet M-Star consistently approximates the DDPM interaction matrix. These results

validate that the Markovian projection preserves reverse dynamics, ensuring robust few-step generation across datasets and diffusion horizons.

### E.4. Quantitative isolation of temporal incoherence

To directly isolate the temporal incoherence hypothesis discussed in Section 3, we measure the variance of $\hat{\mathbf{z}}_0$ predictions across timesteps on ImageNet-64 and ImageNet-128. As noted in the main text, high variance in clean-data prediction degrades sample quality and hinders distillation. Tables 9 and 10 confirm this quantitatively: M-Star reduces $\mathrm{Var}(\hat{\mathbf{z}}_0)$ by $\sim 53\%$ relative to SS-DDPM on both resolutions, and the reduction correlates monotonically with FID.

*Table 9.* Variance of $\hat{\mathbf{z}}_0$ predictions and FID on ImageNet-64. Variance is averaged across all timesteps; lower is better.

| MODEL | $\mathrm{Var}(\hat{\mathbf{z}}_0) \downarrow$ | FID $\downarrow$ (8) | FID $\downarrow$ (1000) |
|---|---|---|---|
| SS-DDPM | $0.312 \pm 0.024$ | $12.45 \pm 0.42$ | $3.87 \pm 0.18$ |
| M-STAR, FIXED $\mathbf{M}_t$ | $0.198 \pm 0.018$ | $4.38 \pm 0.21$ | $1.44 \pm 0.07$ |
| M-STAR, LEARNED $\mathbf{M}_t$ | $\mathbf{0.147 \pm 0.015}$ | $\mathbf{4.12 \pm 0.18}$ | $\mathbf{1.42 \pm 0.06}$ |

*Table 10.* Variance of $\hat{\mathbf{z}}_0$ predictions and FID on ImageNet-128.

| MODEL | $\mathrm{Var}(\hat{\mathbf{z}}_0) \downarrow$ | FID $\downarrow$ (8) | FID $\downarrow$ (1000) |
|---|---|---|---|
| SS-DDPM | $0.334 \pm 0.026$ | $14.21 \pm 0.45$ | $4.12 \pm 0.21$ |
| M-STAR, FIXED $\mathbf{M}_t$ | $0.211 \pm 0.019$ | $3.98 \pm 0.22$ | $1.81 \pm 0.09$ |
| M-STAR, LEARNED $\mathbf{M}_t$ | $\mathbf{0.158 \pm 0.016}$ | $\mathbf{1.89 \pm 0.12}$ | $\mathbf{1.76 \pm 0.08}$ |

To further isolate the effect of $\mathbf{M}_t$, we progressively scale the learned matrix $\mathbf{M}_t^\star$ from 0 (which recovers SS-DDPM) to $2\mathbf{M}_t^\star$. Table 11 reveals a monotonic relation between the scaling, the prediction variance, and FID. The minimum FID occurs at the learned value: too small and the process reverts to SS-DDPM; too large and the global guidance from $\hat{\mathbf{z}}_0$ is overwhelmed.

*Table 11.* $\mathbf{M}_t$-scaling ablation on ImageNet-64. $\mathbf{M}_t^\star$ is the learned matrix. The minimum is achieved exactly at the learned value, matching the theoretical prediction of Section 4.

| $\mathbf{M}_t$ SCALING | $\mathrm{Var}(\hat{\mathbf{z}}_0) \downarrow$ | FID $\downarrow$ (8) |
|---|---|---|
| $0 \times \mathbf{M}_t^\star$ (SS-DDPM) | $0.312 \pm 0.024$ | $12.45 \pm 0.42$ |
| $0.25 \times \mathbf{M}_t^\star$ | $0.264 \pm 0.021$ | $9.87 \pm 0.38$ |
| $0.5 \times \mathbf{M}_t^\star$ | $0.221 \pm 0.019$ | $7.43 \pm 0.31$ |
| $0.75 \times \mathbf{M}_t^\star$ | $0.181 \pm 0.016$ | $5.61 \pm 0.24$ |
| $1.0 \times \mathbf{M}_t^\star$ (M-STAR) | $\mathbf{0.147 \pm 0.015}$ | $\mathbf{4.12 \pm 0.18}$ |
| $1.5 \times \mathbf{M}_t^\star$ | $0.163 \pm 0.017$ | $4.89 \pm 0.21$ |
| $2.0 \times \mathbf{M}_t^\star$ | $0.198 \pm 0.019$ | $6.34 \pm 0.27$ |

### E.5. $\mathbf{M}_t$ ablation on non-Gaussian manifolds

To show that the benefit of the Markovian projection is not specific to the Gaussian case, we ablate $\mathbf{M}_t$ across three synthetic non-Gaussian manifolds. Table 12 reports the KL divergence $D_{\mathrm{KL}}(q(\mathbf{z}_0) \| p_\theta(\mathbf{z}_0))$, estimated with a $k$-NN density estimator on $50\,000$ samples per model. The learned $\mathbf{M}_t$ consistently outperforms both SS-DDPM and the fixed heuristic on vMF, Dirichlet, and Wishart noises. These results extend Table 14 along the $\mathbf{M}_t$ axis.

### E.6. Weather and geodesic benchmarks: setup

We provide additional implementation details for the experiments reported in Section 5.4.

**NASA EOSDIS wildfire (vMF).** Following (Okhotin et al., 2023; NASA EOSDIS, 2020), we use a three-dimensional vMF representation of fire incidence on the unit sphere $\mathbb{S}^2$. The concentration schedule $(\kappa_t)$ is derived analytically from a cosine SNR schedule via Appendix C. Negative log-likelihood (NLL) is the average per-sample log-density on a held-out set. The great-circle distance (GD) is computed in degrees between the predicted mean direction and the ground-truth incident location.

*Table 12.* $\mathbf{M}_t$ ablation on non-Gaussian manifolds. We report the KL divergence $D_{\mathrm{KL}}(q(\mathbf{z}_0) \,\|\, p_\theta(\mathbf{z}_0))$. The learned $\mathbf{M}_t$ outperforms both SS-DDPM and the fixed heuristic across all three manifolds.

| $\mathbf{M}_t$ STRATEGY | vMF KL ↓ | DIRICHLET KL ↓ | WISHART KL ↓ |
|---|---|---|---|
| GAUSSIAN DDPM BASELINE | $0.215 \pm 0.014$ | $0.200 \pm 0.012$ | $0.096 \pm 0.008$ |
| $\mathbf{M}_t = 0$ (SS-DDPM) | $0.024 \pm 0.004$ | $0.011 \pm 0.003$ | $0.037 \pm 0.005$ |
| FIXED $\mathbf{M}_t$ (EQ. 14 ANALOGUE) | $0.010 \pm 0.003$ | $0.006 \pm 0.002$ | $0.025 \pm 0.004$ |
| LEARNED $\mathbf{M}_t$ (M-STAR) | $\mathbf{0.007 \pm 0.002}$ | $\mathbf{0.004 \pm 0.001}$ | $\mathbf{0.021 \pm 0.003}$ |

**CERRA (Ridal et al., 2024).** Pan-European reanalysis, $5.5\,\mathrm{km}$ resolution. We use the $2\,\mathrm{m}$ temperature variable. The training split covers 1985–2018 and evaluation 2019–2020. Following the standard architecture used on ImageNet-64, we use a U-Net with channel multipliers $\{1, 2, 3, 4\}$, AdamW optimizer, batch size 256, and EMA decay 0.9999. CRPS is computed per pixel and averaged over the evaluation period; MMD uses an RBF kernel with bandwidth set by the median heuristic. CRPS values are unscaled; MMD is scaled by $10^3$ for readability.

**HRES-IFS (ECMWF, 2016).** Global forecast system, $9\,\mathrm{km}$ resolution, daily 06Z initialization, $+24\,\mathrm{h}$ lead time. Same training/evaluation protocol as CERRA except the network input is regridded to a $64 \times 64$ patch for compatibility with the ImageNet-64 backbone.

**Step-budget robustness.** For all three datasets, the few-step variant uses the same teacher trained for 1024 steps; the 8-step student is obtained via Progressive Distillation as in Section 4.5. The CRPS drop from 1000 to 8 steps is $\sim 7\%$ for M-Star versus $\sim 40\%$ for SS-DDPM, which is the main quantitative finding of Section 5.4. Table 13 reports the full set of step budgets together with the vMF-KDE baseline on the wildfire dataset.

*Table 13.* Full geodesic and meteorological benchmark, including the 1000-step reference and the vMF-KDE baseline. Lower is better for all metrics.

| DATASET | MODEL | NLL ↓ | GD ↓ (°) | CRPS ↓ | MMD ↓ ($\times 10^{-3}$) | STEPS |
|---|---|---|---|---|---|---|
| WILDFIRE (vMF) | vMF-KDE | $2.84 \pm 0.04$ | $3.21 \pm 0.12$ | – | $4.71 \pm 0.28$ | – |
| WILDFIRE (vMF) | SS-DDPM | $2.58 \pm 0.09$ | $3.02 \pm 0.22$ | – | $4.14 \pm 0.38$ | 8 |
| WILDFIRE (vMF) | **M-Star** | $\mathbf{2.29 \pm 0.06}$ | $\mathbf{2.31 \pm 0.17}$ | – | $\mathbf{2.89 \pm 0.27}$ | 8 |
| CERRA (2 M TEMP.) | SS-DDPM | – | – | $0.412 \pm 0.018$ | $5.24 \pm 0.31$ | 1000 |
| CERRA (2 M TEMP.) | SS-DDPM | – | – | $0.589 \pm 0.024$ | $7.83 \pm 0.44$ | 8 |
| CERRA (2 M TEMP.) | **M-Star** | – | – | $\mathbf{0.374 \pm 0.015}$ | $\mathbf{4.47 \pm 0.26}$ | 1000 |
| CERRA (2 M TEMP.) | **M-Star** | – | – | $\mathbf{0.401 \pm 0.018}$ | $\mathbf{4.98 \pm 0.31}$ | 8 |
| HRES-IFS (2 M TEMP.) | SS-DDPM | – | – | $0.438 \pm 0.020$ | $5.67 \pm 0.35$ | 1000 |
| HRES-IFS (2 M TEMP.) | SS-DDPM | – | – | $0.621 \pm 0.027$ | $8.34 \pm 0.51$ | 8 |
| HRES-IFS (2 M TEMP.) | **M-Star** | – | – | $\mathbf{0.391 \pm 0.016}$ | $\mathbf{4.71 \pm 0.28}$ | 1000 |
| HRES-IFS (2 M TEMP.) | **M-Star** | – | – | $\mathbf{0.418 \pm 0.019}$ | $\mathbf{5.19 \pm 0.33}$ | 8 |

## E.7. Additional results on Dirichlet and Wishart distributions

*Table 14.* **Quantitative Comparison on Synthetic Manifolds.** We report the KL divergence $D_{KL}(q(x_0) \,\|\, p_\theta(x_0))$ for data on the probabilistic simplex (Dirichlet) and the manifold of positive definite matrices (Wishart).

| MODEL | DIRICHLET (KL ↓) | WISHART (KL ↓) |
|---|---|---|
| DDPM (GAUSSIAN) | 0.200 | 0.096 |
| SS-DDPM | 0.011 | 0.037 |
| **M-STAR (OURS)** | **0.004** | **0.021** |

## E.8. ImageNet Samples

To evaluate the visual quality of our 8-step distilled M-Star model, we generate unconditional samples from a variety of ImageNet classes at $128 \times 128$ resolution. Figures 10 and 11 illustrate representative outputs. Across both figures, the model

produces high-fidelity samples with coherent object structure and fine-grained textures, despite the aggressive step reduction from the original teacher model. This demonstrates that the alternating optimization strategy effectively preserves semantic content and diversity in the generated images. Notably, the samples maintain class-specific characteristics and show minimal artifacts, highlighting the robustness of M-Star under few-step generation.

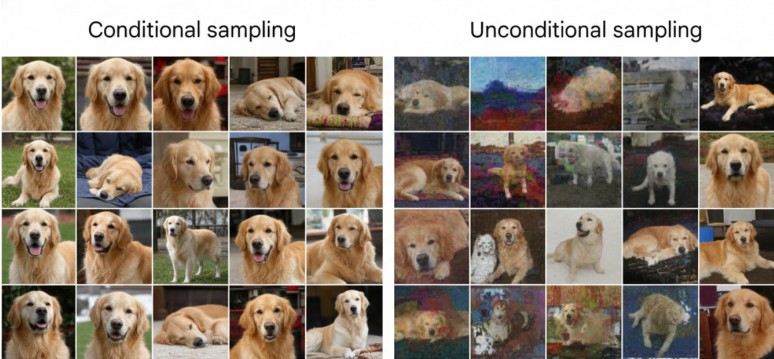

*Figure 10.* Samples from multiple ImageNet classes at $128 \times 128$ resolution, generated using the 8-step distilled model with alternating optimization.

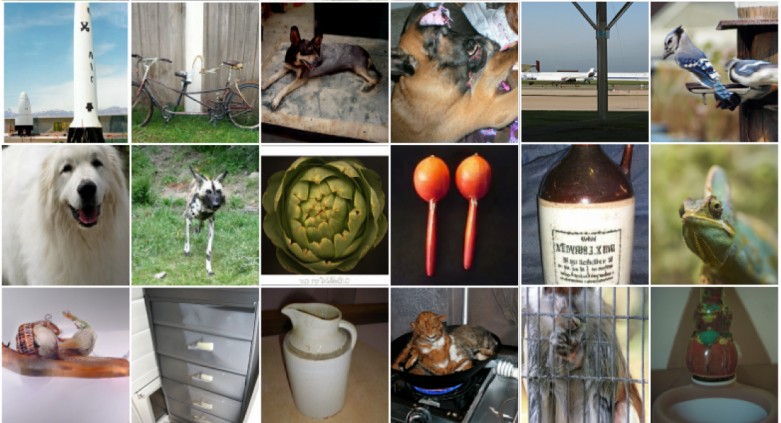

*Figure 11.* Samples from multiple ImageNet classes at $128 \times 128$ resolution, generated using the 8-step distilled model with alternating optimization.

## F. Discrete state spaces

We sketch the extension of M-Star to discrete exponential families, tackling Categorical- and Binomial-based noising processes .

**Categorical noise.** Take $\mathbf{z}_t \in \{e_1, \dots, e_K\} \subseteq \mathbb{R}^K$, the one-hot encoding of a $K$-way categorical variable. Define $q(\mathbf{z}_t \,|\, \mathbf{z}_0)$ as the categorical distribution with mass function

$$q(\mathbf{z}_t = e_i \,|\, \mathbf{z}_0) \propto \exp(\langle \eta_t(\mathbf{z}_0), e_i \rangle), \quad i \in [\![1, K]\!],$$

which fits Assumption 3.2 with sufficient statistic $\psi(\mathbf{z}) = \mathbf{z}$ and reference measure $\nu$ counting on $\{e_i\}$. Setting $\eta_t(\mathbf{z}_0) = \mathbf{A}_t \mathbf{z}_0 + \mathbf{b}_t$ satisfies Assumption 3.3. The kernel construction of (9) then applies verbatim: $\pi_t(\mathbf{z}_{t-1} \,|\, \mathbf{z}_t, \mathbf{z}_0)$ is again categorical with natural parameter (11), and the loss (12) is well-defined since both densities live on the same finite support.

**Binomial noise.** The $\text{Binomial}(n_t, \theta_t(\mathbf{z}_0))$ distribution can be written in canonical form with $\psi(\mathbf{z}) = \mathbf{z}$, $\eta_t(\mathbf{z}_0) = \log(\theta_t(\mathbf{z}_0)/(1 - \theta_t(\mathbf{z}_0)))$, and $f(\mathbf{z}_0) = \mathbf{z}_0$. Provided $\theta_t$ is a smooth function of $\mathbf{z}_0$ (e.g., a logistic transform), Assumption 3.3 holds after a change of variables, and the construction proceeds as in the categorical case.

**Connection to ReMDM.** ReMDM (Lou et al., 2023) is a discrete-space analogue of DDIM that interpolates between an absorbing and a uniform transition kernel. The same Markovian projection viewpoint can be applied to ReMDM: replacing

the absorbing-state hub by an $\hat{\mathbf{z}}_0$ predictor would yield a discrete instantiation of M-Star. We leave a full empirical study of discrete M-Star (and its distillation behaviour) to future work.

