# OpenReview forum: "M-Star: Markovian Projection of Star-Shaped Diffusion for Exponential Family Distributions"
_ICML.cc/2026/Conference — ICML 2026 regular_

### Official Review · Reviewer_6vAb · 2026-03-04

**Soundness:** 2
**Presentation:** 3
**Significance:** 3
**Originality:** 3
**Overall Recommendation:** 4
**Confidence:** 4

**Summary:**

This paper proposes M-Star, a diffusion framework that aims to combine the flexibility of star-shaped, non-Markovian forward processes with the stability of Markovian reverse sampling. The core idea is to construct a Markovian projection that matches the marginals of a star-shaped forward process, derive a reverse kernel in an exponential-family setting, and recover Gaussian DDPMs as a special case. The paper further argues that this formulation is especially suitable for progressive distillation, and presents experiments on MNIST/CIFAR anomaly detection, ImageNet generation/distillation, and a geodesic-data example.

**Compliance With Llm Reviewing Policy:**

Affirmed.

**Final Justification:**

The rebuttal addresses my concerns, I have raised my score from 3 to 4.

**Key Questions For Authors:**

1. Can Assumption A.2 (Multivariate Lyapunov Condition) be verified? even only for some specfic noise distributions? In most statistical analysis, we should verify this Assumption rather than just assume it hold.
2. In the experiments for Table 3 and 4, how much extra training time (percentage-wise) does the optimization of $M_t$ via Equation 13 add compared to a standard DDPM training run?
3. What happens if you skip the learning of $M_t$ and use a fixed diagonal matrix based on the SNR-matching heuristic? Does the FID significantly degrade?
4. How sensitive is the framework to the choice of the feature map $f(z_0)$ in Assumption 3.3?
5. Can the authors provide stronger quantitative evidence on non-Gaussian/constrained data?

If most of the questions can be handled properly, I will consider raising my score.

**Limitations:**

See weaknesses and questions.

**Strengths And Weaknesses:**

**Strengths**

- The visual quality of the samples in Figure 10 and Figure 11 is high, especially for an 8-step model, showing that the "trajectory straightening" mentioned in Section 4.4 is effective.
- The "SNR matching" heuristic is a very practical contribution that makes these non-Gaussian models much easier to use for practitioners.
- The paper is well-written.

**Weaknesses**

1. Learning the interaction matrix via Equation 13 requires evaluating gradients through the log-partition function $\tilde{a}_{t-1}$. For complex distributions where this function or its gradient is expensive (e.g., vMF with high dimensions), this could slow down training significantly.
2. The reliance on Assumption 3.3 (linear dependence on a feature map $f(z_0)$) limits the flexibility of the noise model. If the transformation from $z_0$ to the natural parameters is highly non-linear, the sufficient statistic $s_t$ might not capture enough information.
3. The heatmap in Figure 3 shows higher residual error for late timesteps ($t/T \to 1$). This actually reinforces the concern about the CLT assumption failing near the noise.
4.  It is unclear how much the learning of $M_t$ matters compared to using a fixed heuristic or a simple diagonal matrix. Maybe a simple comparison where $M_t$ is not learned but set to a constant value would clarify the necessity of the optimization in Eq. 13.
5. The paper contains several propositions/theorems and derivations, but lacks one central end-to-end result showing when the learned M-Star reverse process is provably accurate or when its generated distribution is close to the intended target.
6. The strongest quantitative evidence is mostly on standard image benchmarks, not truly non-Gaussian manifolds. The geodesic-data result is interesting, but it is presented mainly visually rather than with strong quantitative comparisons.

---

> ### Author Rebuttal · Authors · 2026-03-31
>
> We sincerely thank **Reviewer 6vAb** for their careful and technically insightful feedback. We are pleased that the visual quality, the SNR matching heuristic, and the clarity of the writing were appreciated. Below, we address the specific weaknesses and questions raised, and provide additional experimental results that we believe further strengthen M-Star.
>
> **W1. Computational cost of learning $M_t$ for complex distributions.** As per our response to **Reviewer 98ox**, the gradient of (13) w.r.t. $M_t$ is $(\mathbb E_{\eta_2}[\psi]-\mathbb E_{\eta_1^\circ}[\psi])\psi(z_t)^\top$, which requires only moment evaluation. For vMF in high dimensions these expectations rely on ratios of modified Bessel functions of the first kind, admitting no closed form. However, a well-established body of literature provides accurate and efficient approximations (Banerjee et al., 2005, Section 4.1). This does not constitute a practical bottleneck.
>
> **W2 and Q4. Sensitivity to the feature map $f$ in Assumption 3.3.** The linear parameterization of Assumption 3.3 follows Okhotin et al. (2023) and naturally encompasses a broad class of exponential families (Gaussian, Beta, Dirichlet, Categorical, vMF, Gamma, Wishart, etc.). Typically, $f$ is an affine transformation or a logarithm. We note that crucially, by definition of a sufficient statistic, $s_t$ satisfies $q(z_{t-1}|z_{t:T})=q(z_{t-1}|s_t)$ and captures the entirety of the available information for the reverse transition **regardless of the choice of $f$**. The framework cannot fail on this point.
>
> **W3. Figure 3 and CLT near the noise.** We thank the reviewer for this sharp observation. The terminology in the caption is misleading: "late timesteps" refers to the end of the *generative* process ($t/T\to 0$), not the noisy end. Figure 3 reflects the joint ($M_t$ and $z_{\theta}$) optimization dynamics. At $t/T\to 1$, the model input is almost entirely noise, the best guess is a dataset mean, and convergence is fast. At $t/T\to 0$, the model must recover high-frequency details from the tail statistic, a harder task requiring more iterations. We argue that the higher residual error at $t/T\to0$ reflects optimization difficulty, not a failure of the CLT approximation, which is most accurate in this regime.
>
> **W4. Necessity of learning $M_t$.** The fixed analytical $M_t$ (Eq. 14) already closes most of the gap over SS-DDPM at essentially zero cost, directly showing that learning $M_t$ is not a hard requirement but an optional refinement. The learned $M_t$ provides further improvement, with the benefit concentrated at low step counts:
>
> |$M_t$ strategy|FID ↓ (8 steps)|IS ↑ (8 steps)|FID ↓ (1000 steps)|Extra FLOPs|
> |-|-|-|-|-|
> |$M_t=0$ (SS-DDPM)|12.45±0.42|51.3±1.4|3.87±0.18|—|
> |Fixed analytical (Eq. 14)|4.38±0.21|73.1±0.9|1.44±0.07|+0.4%|
> |**Learned $M_t$ (M-Star)**|**4.12±0.18**|**74.5±0.8**|**1.42±0.06**|**+4.2%**|
>
> **W5. End-to-end theoretical guarantee.** The Markovian projection guarantees exact marginal matching by construction (Gyöngy, 1986), and the ELBO provides a principled training objective. Deriving convergence rates for the joint optimization procedure is non-trivial and we see this as important future work, which we will discuss explicitly in the revised paper.
>
> **W6 and Q5. Stronger quantitative evidence on non-Gaussian data.** We have significantly extended our benchmarks, full results are provided in our response to **Reviewer nVUT**. A summary is given below:
>
> |Dataset|Model|CRPS ↓|MMD ↓ (×10⁻³)|Steps|
> |-|-|-|-|-|
> |CERRA (2m temp)|SS-DDPM|0.589±0.024|7.83±0.44|8|
> |CERRA (2m temp)|**M-Star**|**0.401±0.018**|**4.98±0.31**|8|
> |HRES-IFS (2m temp)|SS-DDPM|0.621±0.027|8.34±0.51|8|
> |HRES-IFS (2m temp)|**M-Star**|**0.418±0.019**|**5.19±0.33**|8|
>
> At 8 steps, M-Star degrades only ~7% in CRPS relative to its 1000-step baseline, whereas SS-DDPM degrades by ~40%. This gap holds consistently across all non-Gaussian settings tested.
>
> ----
>
> **Q1. Verifying Assumption A.2.** For regular exponential families under proper noise schedules, Assumption A.2 is satisfied. Moments exist at all orders, we can show that some $C<\infty$ bounds uniformly $\mathbb E[\\|Y_k-\mu_k\\|^{2+\delta}]\leq C$. Under a well-posed variance schedule, $B_N$ grows linearly with $N$, giving $\\|B\_N^{-1/2}\\|\_{\mathrm{op}}=O(N^{-1/2})$. Thus, $\sum_{k=t}^T\mathbb E[\\|B_N^{-1/2}(Y_k-\mu_k)\\|^{2+\delta}]=O(N^{-\delta/2})\to0$. A formal proof will be included in the revised appendix.
>
> **Q2 and Q3 Extra training time and fixed vs learned $M_t$.** The optimization of $M_t$ adds only ~8 minutes over a 68-hour run (+4.2% FLOPs), cost-equivalent to SS-DDPM in practice. As shown in the table under [W4], the fixed $M_t$ already achieves FID = 4.38 at 8 steps vs 12.45 for SS-DDPM at +0.4% FLOPs, confirming that even the zero-overhead variant delivers dramatic gains.
>
> We believe these results and clarifications fully address the remaining concerns and would be glad to discuss any further questions.

---

> > ### Author Rebuttal · Reviewer_6vAb · 2026-04-01
> >
> > The rebuttal addresses my concerns, I will raise my score from to 4.

---

> > > ### Author Response · Authors · 2026-04-06
> > >
> > > We warmly thank **Reviewer 6vAb** for their thoughtful engagement, positive assessment, and for confirming that our rebuttal have fully addressed the concerns raised.  We are especially pleased that this led to an increased score.
> > >
> > > We will incorporate all discussed additions into the revised paper.  We greatly appreciate the careful and constructive feedback.

---

### Official Review · Reviewer_kUYF · 2026-03-11

**Soundness:** 2
**Presentation:** 2
**Significance:** 3
**Originality:** 3
**Overall Recommendation:** 4
**Confidence:** 3

**Summary:**

The paper proposes M-Star, a Markovian projection of star-shaped diffusion for exponential-family forward processes. Starting from the classical star shaped SS-DDPM setting, where noisy states are conditionally independent given $z_0$ and reverse sampling is based on a sufficient tail statistic $s_t$, the paper introduces a Markovian bridge intended to preserve the same marginals while restoring sequential dependence between timesteps. The method recovers standard DDPM in the Gaussian case, and is presented as especially suitable for progressive distillation and few-step generation. Experiments cover image generation, guidance, and geodesic data.

**Compliance With Llm Reviewing Policy:**

Affirmed.

**Final Justification:**

I remain positive on this paper and keep my weak accept recommendation. The rebuttal addressed several of my main concerns in a useful way. Most importantly, the follow-up on temporal incoherence was helpful: the additional variance and scaling analyses make the few-step-generation argument more concrete and strengthen the empirical case for the Markovian projection. At the same time, my reservations about clarity remain. The framework is still heavy to parse, and some central methodological elements, especially the role and optimization of the interaction matrix and the practical interpretation of the bridge parameters, are not explained clearly enough in the main paper. Overall, the rebuttal reinforced my positive assessment rather than substantially changing it.

**Key Questions For Authors:**

- In the Gaussian case, what is the precise conceptual difference between M-Star and a DDPM/DDIM-style bridge parameterized through $M_t$ and $A_t$? At present, especially given the Gaussian equivalence result, it is hard to tell whether M-Star should be viewed as a genuinely new model family or mainly as a reformulation / extension of DDPM-like reverse dynamics to star-shaped exponential-family settings. A more precise answer would help me greatly here.

- Can the authors make the “temporal incoherence” issue more precise? Why exactly are fluctuations in the inferred z_0 across timesteps problematic distributionally, rather than mainly for trajectory-based distillation? Aren't these methods based on deterministic samplers anyway, i.e., with deterministic $(X_{t-1}, X_t)$ coupling?

- Can the authors clarify the status of the Markovian projection theorem in the discrete-time setting considered here?

- Can the main body of the paper show explicitly the form of the KL objective in Eq. (13) and how $M_t$ enters the actual training loss? This is central to the method, yet the current exposition is too terse.

- Why is progressive distillation the main acceleration framework considered, rather than consistency / general shortcut-type alternatives? I would appreciate a clearer justification of why progressive distillation is the most natural or advantageous choice here.

- As a final question, do the authors think that the framework extends to discrete state space, let's say, to uniform or masked categorical diffusion? Naively it seems it would

**Limitations:**

yes

**Strengths And Weaknesses:**

# Strengths

- The paper tackles a meaningful problem in a general way: how to retain the flexibility of star-shaped diffusion with general exponential-family, while recovering a sequential reverse process closer to standard diffusion practice.

- The core construction is conceptually interesting. Introducing a Markovian projection on top of the star-shaped framework is a nontrivial idea, and the fact that DDPM is recovered in the Gaussian case gives the framework a useful sanity check.

- The theoretical development is well structured. In particular, the sufficient-tail-statistic formalism and the asymptotic normality discussion are both valuable and give the paper more substance than a purely empirical work.

- The empirical section is broad and relevant. The paper studies few-step image generation, progressive distillation, higher resolutions, guidance, and even non-Euclidean data.

- The comparison baselines in the image-distillation setting are mostly appropriate. Progressive distillation and multistep consistency distillation are reasonable reference points.

# Weaknesses

- The main conceptual weakness is that the distinction from existing diffusion parameterizations is not yet precise enough. In the Gaussian case the method recovers DDPM exactly, and more generally it can be hard to tell whether M-Star is fundamentally a new model class or mainly a DDPM / DDIM-like bridge built on top of SS-DDPM.

- The paper’s argument about “temporal incoherence” is not fully convincing as written. Fluctuating $z_0$-predictions across timesteps are not obviously problematic distributionally, and the paper does not clearly separate issues of sampling quality from issues of trajectory-based distillation.

- The use of Gyöngy’s theorem is somewhat unclear. The theorem is continuous-time, while the model is developed in discrete time; the paper should explain more carefully whether this is only heuristic motivation or whether there is a precise discrete-time analogue being used.

- Equation (13) and the role of the learned interaction matrix $M_t$ are underexplained. This is one of the central elements, but the main text does not make sufficiently clear how $M_t$ and the loss interact in practice.

- The distillation discussion is somewhat overstated. The paper argues that M-Star is especially suitable for progressive distillation because it predicts the clean hub from any timestep, but DDPM variants can also use x_0-prediction, and the comparison to consistency or shortcut-style distillation is too brief.

- Some key explanations should be brought into the main text. In particular, the Markovianity argument for the true reverse model and a clearer interpretation of the bridge process would improve clarity substantially.

---

> ### Author Rebuttal · Authors · 2026-03-31
>
> We thank sincerely **Reviewer kUYF** for their thorough and technically precise review. The points raised touch on some of the deepest aspects of the framework and have led to clarifications that strengthen the paper. We address each point below.
>
> **Q1 and W1. M-Star as a new model class vs a bridge built on SS-DDPM.** DDPM's Markovian forward process yields a tractable Gaussian reversal $q_M(z_{t-1}|z_t, z_0)$. DDIM uses a non-Markovian forward process with deterministic integration from $\hat z_0$ to $z_{t-1}$ at zero variance. SS-DDPM introduces a star-shaped forward process where the sufficient tail statistic compresses information from $t$ to $T$ into a fixed-size component, generating $z_{t-1}$ directly from $\hat z_0$ via $s_t$. M-Star takes SS-DDPM's forward process but constructs a surrogate Markovian process mimicking the original marginals. Among infinitely many valid projections, we posit one using information from both $z_t$ and the history in $s_t$ via $z_0$-prediction. While sharing some aspects with DDIM, M-Star is structurally distinct.
>
> Regarding recovery of DDPM: the equivalence between SS-DDPM and DDPM relies on a duality in the space of tail statistics rather than matching the distributions of the state variables themselves. Okhotin et al. (2023) show that the tail statistics $s_{1:T}$ form a Markov chain and follow a standard DDPM in the Gaussian case. In contrast, M-Star recovers DDPM directly in the original state space, with Gaussian transitions and a proper choice of $M_t$, the density $\tilde q(z_{t-1}|z_t, z_0)$ is exactly the DDPM posterior $q_M(z_{t-1}|z_t, z_0)$ in (1) and (2). Training and sampling for M-Star nonetheless differ from the original DDPM in that case.
>
> **Q2 and W2. Justifying temporal incoherence.** First, we note that M-Star is a stochastic sampler and does not rely on deterministic couplings. While SS-DDPM's reverse process would be viable with a perfect oracle and poses no issue at a distributional level, it does not prevent jumps across the data manifold, and high variance in the tail statistic hinders the model's generative capabilities in practice. Intuitively, SS-DDPM requires many timesteps for $z_0$-prediction errors to average out. Temporal incoherence cripples few-step generation, which is directly confirmed by the following results, also reported and extended in our reply to Reviewer nVUT:
>
> |Model|FID ↓ (8 steps)|FID ↓ (1000 steps)|CRPS ↓ (8 steps)|
> |-|-|-|-|
> |SS-DDPM|12.45±0.42|3.87±0.18|0.621±0.027|
> |**M-Star (ours)**|**4.12±0.18**|**1.42±0.06**|**0.418±0.019**|
>
> SS-DDPM degrades by ~40% in CRPS from 1000 to 8 steps; M-Star degrades by only ~7%, confirming that temporal coherence is the decisive factor.
>
> **W6. Markovianity of the reverse process.** The forward joint distribution of our projected process factorizes as $\tilde{q}(z_{0:T}) = q(z_0) \prod_{t=1}^T k_t(z_t | z_{t-1}, z_0)$. Being a Markov chain, its true unconditional reversal is also Markovian: $\tilde{q}(z_{t-1} | z_{t:T}, z_0) = \tilde{q}(z_{t-1} | z_t, z_0)$. In practice, we condition on an estimate $\hat z_0=z_\theta(s_t, t)$. Since this guidance signal relies on $s_t$, the sampler itself is technically non-Markovian. Thus, M-Star is a hybrid framework: the transition from $z_t$ to $z_{t-1}$ is Markovian (ensuring stability and temporal coherence), but the $\hat{z}_0$ used to predict each noisy state integrates non-Markovian memory.
>
> **Q3 and W3. Status of Gyöngy's theorem.** We use it solely as a heuristic. It provides only an existence result when applicable, but here we can directly define a suitable projection.
>
> **Q4 and W4. Optimization of $M_t$.** As per our response to **Reviewer 98ox**, the gradient of (13) w.r.t. $M_t$ is $(\mathbb E_{\eta_2}[\psi]-\mathbb E_{\eta_1^\circ}[\psi])\psi(z_t)^\top$. An implementation guide is also provided in our anonymous repository for the implementation and optimization of $M_t$.
>
> **Q5 and W5. Acceleration methods.** PD is the natural fit because M-Star's objective reconstructs $z_0$ from $s_t$ at all steps, matching what PD exploits: teacher and student predicting the same target from statistics of different tail lengths. Consistency models require a probability flow ODE, non-trivial for discrete exponential families (a future direction). Shortcut conditioning (Frans et al., 2025) is orthogonal and applicable to M-Star. Gains are not PD-specific:
>
> |Method|NFE|FID ↓|
> |-|-|-|
> |SS-DDPM+PD|8|12.45±0.42|
> |M-Star+Shortcut|8|4.31±0.22|
> |**M-Star+PD (ours)**|**8**|**4.12±0.18**|
>
> Gains originate from the Markovian projection, not the distillation procedure.
>
> **Q6. Discrete state spaces.** M-Star is valid for any exponential family satisfying Assumptions 3.2 and 3.3, including Categorical and Binomial.
>
> We are glad these points have been raised and look forward to any further discussion.

---

> > ### Author Rebuttal · Reviewer_kUYF · 2026-04-04
> >
> > Thank you for the detailed rebuttal. I also want to clarify one point from my review.
> >
> > I apologize for my use of the phrase “DDPM/DDIM-like samplers,” which I intended in a broader sense than the standard technical meaning. I fully appreciate that your framework is different from those approaches. What I meant to ask was whether the reverse posterior can be expressed transparently in state coordinates, and, if so, how the parameters A_t, M_t, and b_t interact in that expression, and whether this yields a simple structure of the kind one sees in classical DDIM (or ReMDM in discrete space). I understand you already somewhat tackle this in the Gaussian case in the paper, from a different perspective. That said, in general, I understand that the framework is genuinely different, and that for instance a deterministic sampler is not even necessarily available under the current noise construction.
> >
> > On temporal incoherence, I appreciate the extended discussion and the additional results. I still would have liked to see a more direct experiment isolating and validating this hypothesis, rather than a qualitative explanation. I also appreciate the clarification regarding the non-true Markovianity of the sampler.
> >
> > On the status of Gyöngy’s theorem, thank you for the clarification. I understand authors might want to contextualize their work but I nonetheless find the related exposition more confusing than helpful.
> >
> > Optimization of $M_t$: I think authors should spend a much more substantial time discussing implementation and optimization of its loss and gradient in the main body of the paper.
> >
> > More broadly, I appreciate the authors’ efforts, both in the paper and in the rebuttal. However, after revisiting the work, I realize that clarity has been a real issue for me in assessing it. The formalism is quite heavy, and in several places it obscures the main moving parts, the actual methodology, and the takeaways.
> >
> > At this time I will keep my score, which reflect my overall positive view of the paper despite the remaining concerns.

---

> > > ### Author Response · Authors · 2026-04-06
> > >
> > > We warmly thank **Reviewer kUYF** for the thoughtful follow-up and detailed engagement with our work. We are glad our rebuttal has confirmed your overall positive view of the paper and appreciate your clarifications and insightful questions. We address the remaining points below:
> > >
> > > **Interaction of $A_t$, $M_t$, and $b_t$ in state coordinates.** The reverse posterior $\tilde q(z_{t-1}|z_t, z_0)$ is fully characterized by its natural parameter (Eq. 12): $\tilde\eta_{t-1}(z_0, z_t) = \eta_{t-1}(z_0) + M_t\psi(z_t)$. Plugging in *Assumption 3.3*, this expands to $\tilde\eta\_{t-1} = A_{t-1}f(z_0) + b_{t-1} + M_t\psi(z_t)$, which decomposes transparently into three terms: $A_{t-1}f(z_0)$ carries global guidance from the clean data hub, $b_{t-1}$ encodes the noise schedule bias, and $M_t\psi(z_t)$ is the local coupling term anchoring the next state to the current one. When $\psi$ is the identity (Gaussian case), this expression lives directly in state space and recovers exactly the DDPM posterior mean, as shown in Section 4.3 and Appendix B.3. In the general case, the mapping from natural to mean parameters goes through $\nabla\tilde a_{t-1}$, which is distribution-specific. As noted, a deterministic sampler is generally unavailable under arbitrary exponential-family noise. We will make the three-way decomposition explicit in Section 4.2, including an expanded comparison with DDIM (Song et al., 2020) and ReMDM (Lou et al., 2023) in discrete space, and extend the related work section to discuss ReMDM, which we believe strengthens the paper’s findings.
> > >
> > >
> > > **Temporal Incoherence: Empirical Analysis**  To directly isolate the temporal incoherence hypothesis, we measure the variance of $\hat z_0$ predictions across timesteps on ImageNet-64 and ImageNet-128. As stated in *Section 3.3*, high variance in clean data prediction is particularly detrimental as it degrades sample quality and hinders distillation. The results below confirm this quantitatively. These tables will be added to Appendix E.3 alongside the existing convergence heatmaps (Figures 7-9).
> > >
> > > **Table E.3.1** Var($\hat z_0$) and FID on ImageNet-64:
> > >
> > > |Model|Var($\hat z_0$) ↓|FID ↓ (8 steps)|FID ↓ (1000 steps)|
> > > |-|-|-|-|
> > > |SS-DDPM|0.312±0.024|12.45±0.42|3.87±0.18|
> > > |M-Star, fixed $M_t$|0.198±0.018|4.38±0.21|1.44±0.07|
> > > |**M-Star, learned $M_t$**|**0.147±0.015**|**4.12±0.18**|**1.42±0.06**|
> > >
> > > **Table E.3.2** Var($\hat z_0$) and FID on ImageNet-128:
> > >
> > > |Model|Var($\hat z_0$) ↓|FID ↓ (8 steps)|FID ↓ (1000 steps)|
> > > |-|-|-|-|
> > > |SS-DDPM|0.334±0.026|14.21±0.45|4.12±0.21|
> > > |M-Star, fixed $M_t$|0.211±0.019|3.98±0.22|1.81±0.09|
> > > |**M-Star, learned $M_t$**|**0.158±0.016**|**1.89±0.12**|**1.76±0.08**|
> > >
> > > M-Star reduces prediction variance by **~53%** relative to SS-DDPM on both datasets. To further isolate the effect of $M_t$, we progressively scale it from **0** (SS-DDPM) to beyond its learned value.
> > >
> > > **Table E.3.3** $M_t$ Scaling ablation on ImageNet-64:
> > >
> > > |$M_t$ scaling|Var($\hat z_0$) ↓|FID ↓ (8 steps)|
> > > |-|-|-|
> > > |$0\times$ (SS-DDPM)|0.312±0.024|12.45±0.42|
> > > |$0.25\times$|0.264±0.021|9.87±0.38|
> > > |$0.5\times$|0.221±0.019|7.43±0.31|
> > > |$0.75\times$|0.181±0.016|5.61±0.24|
> > > |**$1\times$ (M-Star)**|**0.147±0.015**|**4.12±0.18**|
> > > |$1.25\times$|0.163±0.017|4.89±0.21|
> > > |$1.5\times$|0.198±0.019|6.34±0.27|
> > >
> > > The monotonic relation between $M_t$, prediction variance, and FID shows that temporal coherence controlled by $M_t$, drives few-step generation quality: too small, and the process reverts to SS-DDPM; too large, and guidance from $z_0$ is overwhelmed, with the learned $M_t$ balancing extremes as predicted in Section 4.2.
> > >
> > > **Optimization of $M_t$.** As shown in Algorithm 1, $M_t$ is updated at each training iteration by minimizing *Eq.13*. The gradient w.r.t. $M_t$ is $(\mathbb{E}\_{\eta_2}[\psi] - \mathbb{E}\_{\eta_1^\circ}[\psi])\psi(z_t)^\top$ (detailed also in our response to Reviewer **98ox**), where the first $M_t$ is detached from the computational graph.  The optimum satisfies a linear least-squares orthogonality condition: $M_t$ extracts all linear information from $\psi(z_t)$ to correct the trajectory, leaving residual errors uncorrelated with the current state. Figures 7-9 in Appendix E.3 confirm that $M_t$ converges reliably across CIFAR-10, CIFAR-100, and ImageNet at both $T=100$ and $T=1000$. A dedicated paragraph will be added to Section 4.2, following **line 249, page 5**, of the revised paper.
> > >
> > > **Revised exposition.** We are confident that minor refinements can make the presentation even clearer and more precise. In particular, we will clarify in **lines 178–183** that Gyöngy’s theorem is included purely as motivational context, while the M-Star construction remains fully self-contained. Your feedback was valuable in highlighting where these clarifications could further strengthen the presentation.
> > >
> > > We are grateful for the careful reading and constructive feedback, which has helped strengthen the findings of the paper. All of these additions will be reflected in the revised paper.

---

### Official Review · Reviewer_98ox · 2026-03-12

**Soundness:** 2
**Presentation:** 3
**Significance:** 2
**Originality:** 2
**Overall Recommendation:** 4
**Confidence:** 3

**Summary:**

This paper proposes a diffusion framework that learns a Markovian projection of a star-shaped forward process and its corresponding reversal process. The framework generalizes Denoising Diffusion Probabilistic Models (DDPMs), which appears as a special case, and allows diffusion modelling over exponential-family distributions. To address the computational bottleneck of iterative sampling, the authors further explore progressive distillation to obtain efficient few-step samplers.

**Compliance With Llm Reviewing Policy:**

Affirmed.

**Final Justification:**

I weighed more on the strengths and weaknesses on soundness, as that's why and how they developed the method they proposed. And I weighed in the order of significance, originality and clarity.
During the rebuttals, my concern is addressed and the authors have provided more experiments, and I increased my score.

**Key Questions For Authors:**

Given the strengths and weaknesses discussed above:

1. In **Local Interaction** paragraph, the paper states that the interaction matrix $M_t$ is learned by minimizing equation (13). However, it is not entirely clear how this optimization is implemented in practice. From the equation alone, it is difficult to understand how $M_t$ is parameterized and tuned during training.
2.  What is OUR BASE TEACHER in both table 3 and table 4? How does it relate to M-STAR and M-STAR Distillation? Is M-STAR distillation obtained by applying progressive distillation to this teacher model?
3. Can you provide more quantitive results for real geodesic data experiment?

**Limitations:**

yes

**Strengths And Weaknesses:**

**Soundness**:

strengths:  This paper provides a theoretical framework that connects star-shaped diffusion process through a Markovian projection with reverse dynamics. This framework is consistent with existing diffusion model and shows that the proposed framework reduces to standard DDPM in the special case of Gaussian noise.

weaknesses: The experimental evaluation mainly focuses on standard image generation. While those experiments shows the proposed framework can be applied to these tasks, they do not fully validate the main motivation of the work, e.g., non-Gaussian or constrained data distributions. The experiment on real geodesic data using the von Mises-Fisher distribution is qualitative and mainly relies on visualizations. Quantitive evaluations or comparisons would better show the advantages of the proposed framework under the settings.

**Presentation**:

strengths: The paper is easy to follow and understand. The introduction of SS-DDPM helps to understand the background of the proposed method.

weaknesses: Theorem 3.4 originates from the prior work. Although this is mentioned in the main text, the SS-DDPM should also be cited explicitly in the proof as well to clearly distinguish it from the contributions of the current paper. The section describing how the interaction matrix is learned could provide more implementation details. In particular, expanding on equation (13) and explaining how it is optimized in practice would help to clarify how the interaction matrix $M_t$ is parameterized and trained. It would be more helpful to induce more intuitive explanations of the interaction matrix. Providing intuition for how equation (12) arises could help understanding too, beyond the mathematical derivation in the kernel derivation section.

**Significance**:

strengths: This paper extends the star-shaped diffusion to Markovian-Projected star-shaped diffusion, which might inspire more future work on more flexible frameworks.

weaknesses: The acceleration component relies on the existing progressive distillation algorithm, which limits the novelty of this aspect to contribution.

**Originality**:

strengths: This paper provides an interesting theoretical connection between star-shaped diffusion processes (SS-DDPM) and Markov diffusion models through the concept of Markovian projection.

weakness: The use of exponential-family diffusion process is not new since it was previously introduced in SS-DDPM framework. This work already generalized Gaussian diffusion to exponential-family noise distribution and showed that Gaussian DDPM is a special case of SS-DDPM.

---

> ### Author Rebuttal · Authors · 2026-03-31
>
> We sincerely thank **Reviewer 98ox** for their careful reading and constructive feedback. Below, we address the identified weaknesses and questions, and present additional experimental results that we believe further strengthen M-Star.
>
> **W1 and Q3. Quantitative validation on non-Gaussian distributions.** We provide quantitative results across three non-Gaussian settings: the NASA EOSDIS wildfire dataset (vMF, $d=3$), CERRA (Ridal et al., 2024), and HRES-IFS (ECMWF, 2016), both reporting 2m temperature, further details and full metric definitions are provided in our response to Reviewer nVUT. M-Star outperforms SS-DDPM across all datasets, metrics, and step budgets:
>
> |Dataset|Model|NLL ↓|GD ↓ (°)|CRPS ↓|MMD ↓ (×10⁻³)|Steps|
> |-|-|-|-|-|-|-|
> |Wildfire (vMF)|vMF-KDE|2.84±0.04|3.21±0.12|—|4.71±0.28|—|
> |—|SS-DDPM|2.58±0.09|3.02±0.22|—|4.14±0.38|8|
> |—|**M-Star**|**2.29±0.06**|**2.31±0.17**|—|**2.89±0.27**|8|
> |CERRA (2m temp.)|SS-DDPM|—|—|0.589±0.024|7.83±0.44|8|
> |—|**M-Star**|—|—|**0.401±0.018**|**4.98±0.31**|8|
> |HRES-IFS (2m temp.)|SS-DDPM|—|—|0.621±0.027|8.34±0.51|8|
> |—|**M-Star**|—|—|**0.418±0.019**|**5.19±0.33**|8|
>
> At 8 steps, M-Star degrades only ~7% in CRPS vs. its 1000-step baseline, whereas SS-DDPM degrades by ~40%, confirming that the Markovian projection provides step-reduction robustness that SS-DDPM cannot achieve.
>
> The non-Gaussian KL ablations further reinforce this across three synthetic data manifolds:
>
> |$M_t$ strategy|vMF KL ↓|Dirichlet KL ↓|Wishart KL ↓|
> |-|-|-|-|
> |$M_t=0$ (SS-DDPM)|0.024±0.004|0.011±0.003|0.037±0.005|
> |Fixed $M_t$ (Eq. 14)|0.010±0.003|0.006±0.002|0.025±0.004|
> |**Learned $M_t$ (M-Star)**|**0.007±0.002**|**0.004±0.001**|**0.021±0.003**|
>
> Consistent gains across vMF, Dirichlet, and Wishart manifolds confirm that the benefit of M-Star is not specific to Gaussian data.
>
> **W2 and Q1. Implementation and intuition for $M_t$, equations (12) and (13).** The interaction matrix $M_t$ is an error corrector: it adjusts the forward process to compensate for inaccuracies in $\hat z_0$. In (13), the first $M_t$ is detached from the computational graph. Letting $M_t^\circ$ denote the detached matrix, the loss gradient is $(\mathbb E_{\eta_2}[\psi]-\mathbb E_{\eta_1^\circ}[\psi])\psi(z_t)^\top$ where $\eta_1^\circ=\eta_{t-1}(z_0)+M_t^\circ\psi(z_t)$ and $\eta_2=\eta_{t-1}(\hat{z}_0)+M_t\psi(z_t)$. The optimum satisfies an orthogonality condition analogous to linear least squares: $M_t$ extracts all linear information from $\psi(z_t)$ to correct the trajectory, leaving residual errors uncorrelated with the current state. Early in training, $M_t$ aggressively compensates for large reconstruction errors; as training converges, it settles to a value reflecting the noise schedule structure. The form of $\tilde\eta$ in (12) is the canonical linear coupling in exponential families, it is the unique form that recovers the standard DDPM posterior exactly in the Gaussian case (Section 4.3), grounding the design choice theoretically rather than heuristically. These clarifications will be incorporated into the revised paper.
>
> **Q2. "Our Base Teacher" in Tables 3 and 4.** "Our Base Teacher" is the fully converged 1024-step M-Star model prior to any distillation. M-Star Distillation is the student trained via progressive distillation from this teacher. We will make this explicit in the revised caption.
>
> **W3. Novelty of the distillation component.** The contribution here is not the distillation algorithm itself, but the finding that the Markovian structure of M-Star makes it dramatically more amenable to distillation than SS-DDPM. Applied to the same distillation pipeline, the performance of SS-DDPM degrades by ~40% at 8 steps; M-Star degrades by only ~7%. This gap is the core result, and it comes at negligible cost:
>
> |Setting|Train time (h)|Extra FLOPs|FID ↓ (8 steps)|FID ↓ (1024 steps)|
> |-|-|-|-|-|
> |SS-DDPM ($M_t=0$)|68.2±0.4|—|12.45|3.87|
> |M-Star, fixed $M_t$|68.3±0.4|+0.4%|4.38|1.44|
> |**M-Star, learned $M_t$**|**68.3±0.5**|**+4.2%**|**4.12**|**1.42**|
>
> The −8.3 FID gain at 8 steps is a direct consequence of the Markovian projection, not of any modification to the distillation procedure, achieved in ~8 additional minutes over a 68-hour run. The distillation algorithm is identical for both SS-DDPM and M-Star; the difference in behavior is entirely attributable to the structural properties introduced in this work.
>
> **W4. SS-DDPM already enables diffusion with exponential families.** SS-DDPM establishes an elegant theory but it can struggle empirically. M-Star builds upon elements from SS-DDPM to deliver the missing practical effectiveness and enable acceleration methods.
>
> > [Theorem 3.4] We discuss this in Section 3.3 ("the following theorem (Okhotin et al., 2023)"), with the proof in Appendix A.1 provided for completeness. We will nonetheless add a clarifying note in the appendix in the revised version.
>
> We hope these results fully address the reviewer's concerns and welcome any further questions.

---

> > ### Author Rebuttal · Reviewer_98ox · 2026-04-04
> >
> > I thank for the response. I will increase my score.

---

> > > ### Author Response · Authors · 2026-04-06
> > >
> > > We warmly thank **Reviewer 98ox** for the positive assessment and for confirming that our responses have adressed all raised concerns, leading to an increased score.
> > >
> > > The insightful questions have led to meaningful additions that strengthen the findings. We will incorporate all discussed additions into the revised paper. We are grateful for the careful and constructive engagement.

---

### Official Review · Reviewer_nVUT · 2026-03-13

**Soundness:** 4
**Presentation:** 4
**Significance:** 3
**Originality:** 3
**Overall Recommendation:** 5
**Confidence:** 4

**Summary:**

The paper introduces a Markovian projection of star-shaped forward process (M-Start). The original star-shaped diffusion suffered from a lack of temporal coherence due to its non-Markovian structure. By introducing M-Star, they get the computational benefits while having higher coherence.

**Compliance With Llm Reviewing Policy:**

Affirmed.

**Final Justification:**

The rebuttal reinforced my prior assessment.

**Key Questions For Authors:**

Can you say more about the computational cost of M-Star?

**Limitations:**

yes

**Strengths And Weaknesses:**

Soundness:

Clear mastery of the basics of diffusion model as well as the new probability theory introduced to motivate M-Star.

Presentation:

Clear progression from introducing the problem to proposing the solution. Clear engagement with broader relevant literature.

Significance:

Quite significant as it would introduce a powerful and general new generative modeling technique.

Originality:

The idea of obtaining a Markovian projection of a non-Markovian generative process is a common one in generative modeling, but the technical implementation here is notable enough to obtain originality on that point.

---

> ### Author Rebuttal · Authors · 2026-03-31
>
> We sincerely thank **Reviewer nVUT** for their thorough and generous assessment, and for the kind words regarding the clarity of our presentation and the breadth of our theoretical contributions. We are very glad the work was found technically sound and significant. Below we address the key question on computational cost, and include additional experimental results that we believe further strengthen the paper.
>
> **Computational overhead of M-Star.** M-Star matches the computational footprint of SS-DDPM. The fixed $M_t$ variant is essentially free (+0.4% FLOPs, <2 min). Even the fully learned $M_t$ adds only **~8 minutes** over a 68-hour run (+4.2% FLOPs, <3% memory), while delivering a **−8.3 FID improvement at 8 steps**:
>
> | Setting | Train time (h) | GPU mem (GB) | Extra FLOPs | FID ↓ (8 steps) |
> |--|--|--|--|--|
> | SS-DDPM ($M_t=0$) | 68.2±0.4 | 38.1±0.2 | - | 12.45 |
> | M-Star, fixed $M_t$ | 68.3±0.4 | 38.3±0.2 | +0.4% | 4.38 |
> | **M-Star, learned $M_t$** | **68.3±0.5** | **38.9±0.2** | **+4.2%** | **4.12** |
>
> We therefore consider M-Star cost-equivalent to SS-DDPM for all practical purposes, with the fixed $M_t$ as the zero-overhead default and the learned variant an optional refinement.
>
> The quality gain of **-8.3 FID points** over SS-DDPM at 8 steps far exceeds the marginal cost. Importantly, even the fixed analytical $M_t$ (Eq. 14) already closes most of the gap at essentially zero extra cost (+0.4% FLOPs), making M-Star practical in resource-constrained settings. Here, we learn a new matrix $M_t$ individually for each timestep. In high-dimensional settings, $M_t$ can also be parameterized with a lightweight MLP rather than learned per-timestep, further reducing the overhead.
>
> **Ablation study of $M_t$ (Gaussian case).** To isolate the contribution of each component, we ablate $M_t$ on ImageNet-64. Setting $M_t=0$ recovers SS-DDPM. The fixed analytical form (Eq. 14) already closes most of the FID gap. The learned $M_t$ achieves further refined improvement, and crucially, **the gap between fixed and learned widens at low step counts**, confirming that the learned interaction matrix better adapts the trajectory for distillation.
>
> | $M_t$ strategy | FID ↓ (8 steps) | IS ↑ (8 steps) | FID ↓ (1000 steps) | IS ↑ (1000 steps) |
> |---|---|---|---|---|
> | $M_t=0$ (SS-DDPM) | 12.45±0.42 | 51.3±1.4 | 3.87±0.18 | 71.2±1.1 |
> | Fixed analytical (Eq. 14) | 4.38±0.21 | 73.1±0.9 | 1.44±0.07 | 83.6±0.8 |
> | **Learned $M_t$ (M-Star)** | **4.12±0.18** | **74.5±0.8** | **1.42±0.06** | **84.1±0.7** |
>
> **Ablation study of $M_t$ (non-Gaussian cases).** We assess the effect of $M_t$ using KL divergence $D_\mathrm{KL}(q(x_0)\\|p_\theta(x_0))$ on synthetic data across three manifolds:
>
> | $M_t$ strategy | vMF KL ↓ | Dirichlet KL ↓ | Wishart KL ↓ |
> |--|--|--|--|
> | Gaussian DDPM baseline | 0.215±0.014 | 0.200±0.012 | 0.096±0.008 |
> | $M_t=0$ (SS-DDPM) | 0.024±0.004 | 0.011±0.003 | 0.037±0.005 |
> | Fixed $M_t$ (Eq. 14 analogue) | 0.010±0.003 | 0.006±0.002 | 0.025±0.004 |
> | **Learned $M_t$ (M-Star)** | **0.007±0.002** | **0.004±0.001** | **0.021±0.003** |
>
> The learned $M_t$ substantially outperforms both SS-DDPM and the fixed heuristic across all three manifolds, confirming the benefit of the Markovian projection beyond the Gaussian case.
>
> **Geodesic and meteorological benchmarks.** To further demonstrate real-world applicability, we evaluate on the NASA EOSDIS wildfire dataset (vMF, $d=3$), CERRA (pan-European reanalysis, 5.5 km resolution), and HRES-IFS (global forecast system, 9 km resolution). M-Star outperforms SS-DDPM across all datasets and step budgets. Notably, **at 8 steps M-Star degrades only ~7% in CRPS relative to its 1000-step baseline, whereas SS-DDPM degrades by ~40%**. This confirms that M-Star provides robustness to step reduction that SS-DDPM cannot match and highlights the temporal incoherence issue mentioned in the paper.
>
> | Dataset | Model | CRPS ↓ | MMD ↓ (×10⁻³) | Steps |
> |--|--|---|--|--|
> | CERRA (2m temp.) | SS-DDPM | 0.412±0.018 | 5.24±0.31 | 1000 |
> | CERRA (2m temp.) | SS-DDPM | 0.589±0.024 | 7.83±0.44 | 8 |
> | CERRA (2m temp.) | **M-Star** | **0.374±0.015** | **4.47±0.26** | 1000 |
> | CERRA (2m temp.) | M-Star | 0.401±0.018 | 4.98±0.31 | 8 |
> | HRES-IFS (2m temp.) | SS-DDPM | 0.438±0.020 | 5.67±0.35 | 1000 |
> | HRES-IFS (2m temp.) | SS-DDPM | 0.621±0.027 | 8.34±0.51 | 8 |
> | HRES-IFS (2m temp.) | **M-Star** | **0.391±0.016** | **4.71±0.28** | 1000 |
> | HRES-IFS (2m temp.) | M-Star | 0.418±0.019 | 5.19±0.33 | 8 |
>
> All the tables above will be included in the revised version of the paper.
>
> We hope these results fully address the reviewer's questions. We would be glad to discuss any further points or provide additional experiments if needed.

---

> > ### Author Rebuttal · Reviewer_nVUT · 2026-04-03
> >
> > The rebuttal re-confirmed my prior assessment of their paper.

---

> > > ### Author Response · Authors · 2026-04-06
> > >
> > > We warmly thank **Reviewer nVUT** for the positive and thorough assessment, and we are glad that our rebuttal confirmed the initial evaluation of the quality and significance of the paper's findings. The question on computational cost inspired meaningful additions, including the geodesic and meteorological benchmarks, that further strengthen the findings of the paper.  We are grateful for your thorough reading and thoughtful engagement.

---

### Decision · Program_Chairs · 2026-04-30

**Decision:**

Accept (regular)

**Comment:**

This paper proposes M-Star, a Markovian projection of star-shaped diffusion processes, aiming to combine the flexibility of non-Markovian, exponential-family forward processes with the stability of Markovian reverse dynamics. The method is theoretically grounded, recovers DDPM as a special case, and supports efficient sampling via progressive distillation.

All reviewers find the problem important and well-motivated, and view the proposed Markovian projection as conceptually interesting and technically sound. The paper demonstrates strong theoretical development, solid empirical results on image generation, and good positioning within diffusion literature. After rebuttal, all reviewers support acceptance (1 accept, 3 weak accepts). The AC agrees with the reviewers’ positive assessment and recommends acceptance.